# Mechanistic insights into a TIMP3-sensitive pathway constitutively engaged in the regulation of cerebral hemodynamics

Carmen Capone[1,2], Fabrice Dabertrand[3,4], Celine Baron-Menguy[1,2], Athena Chalaris[5,6], Lamia Ghezali[1,2], Valérie Domenga-Denier[1,2], Stefanie Schmidt[5,6], Clément Huneau[1,2], Stefan Rose-John[5,6], Mark T Nelson[3,4,7], Anne Joutel[1,2]*

[1]Genetics and Pathogenesis of Cerebrovascular Diseases, INSERM, U1161, Université Paris Diderot, Sorbonne Paris Cité, UMRS 1161, Paris, France; [2]DHU NeuroVasc, Sorbonne Paris Cité, Paris, France; [3]Department of Pharmacology, University of Vermont, Burlington, United States; [4]College of Medicine, University of Vermont, United States; [5]Institute of Biochemistry, Christian Albrechts University, Kiel, Germany; [6]Medical Faculty, Christian Albrechts University, Kiel, Germany; [7]Institute of Cardiovascular Sciences, University of Manchester, Manchester, United Kingdom

*For correspondence: anne. joutel@inserm.fr

Competing interests: The authors declare that no competing interests exist.

**Abstract** Cerebral small vessel disease (SVD) is a leading cause of stroke and dementia. CADASIL, an inherited SVD, alters cerebral artery function, compromising blood flow to the working brain. TIMP3 (tissue inhibitor of metalloproteinase 3) accumulation in the vascular extracellular matrix in CADASIL is a key contributor to cerebrovascular dysfunction. However, the linkage between elevated TIMP3 and compromised cerebral blood flow (CBF) remains unknown. Here, we show that TIMP3 acts through inhibition of the metalloprotease ADAM17 and HB-EGF to regulate cerebral arterial tone and blood flow responses. In a clinically relevant CADASIL mouse model, we show that exogenous ADAM17 or HB-EGF restores cerebral arterial tone and blood flow responses, and identify upregulated voltage-dependent potassium channel ($K_V$) number in cerebral arterial myocytes as a heretofore-unrecognized downstream effector of TIMP3-induced deficits. These results support the concept that the balance of TIMP3 and ADAM17 activity modulates CBF through regulation of myocyte $K_V$ channel number.

## Introduction

Small vessel disease (SVD) of the brain is a leading cause of stroke and age-related cognitive decline and disability for which there are currently no treatments (*Pantoni, 2010*). Our limited understanding of the pathogenesis of cerebral SVD is a major obstacle to the development of treatments. Monogenic forms of these diseases, such as CADASIL (Cerebral Autosomal Dominant Arteriopathy with Subcortical Infarcts and Leukoencephalopathy), provide a window into the mechanism underlying much more common, but largely indistinguishable, sporadic forms of SVD (*Joutel and Faraci, 2014*). CADASIL, the most common hereditary cerebral SVD, is caused by dominant mutations in the NOTCH3 receptor that stereotypically lead to the extracellular deposition of NOTCH3 ectodomain (Notch3$^{ECD}$) and aggregates of other proteins on vessels (*Joutel et al., 2000*; *Chabriat et al., 2009*; *Monet-Leprêtre et al., 2013*). A deficit in cerebral blood flow (CBF) hemodynamics is an early

**eLife digest** There are currently no effective treatments or cures for small blood vessel diseases of the brain, which lead to strokes and subsequent decreases in mental abilities. Normally, smooth muscle cells that surround the vessels relax or contract to regulate blood flow and ensure the right amount of oxygen and nutrients reaches the different regions of the brain. In a syndrome called CADASIL, which is the most common form of inherited small vessel disease, a genetic mutation causes the smooth muscle cells to weaken over time.

The accumulation of several proteins – including one called TIMP3 – around the smooth muscle cells plays a key role in the smooth muscle cell weakening seen in CADASIL. Capone et al. have now studied mice that display the symptoms of CADASIL to investigate how TIMP3 decreases blood flow through blood vessels in the brain. This revealed that TIMP3 inactivates another protein called ADAM17. The latter protein is normally responsible for starting a signaling pathway that helps smooth muscle cells to regulate blood flow according to the needs of the brain cells. Artificially adding more ADAM17 to the brains of the CADASIL mice reduced their symptoms of small vessel disease.

Using smooth muscle cells freshly isolated from the brains of CADASIL mice, Capone et al. also demonstrated that abnormal TIMP3-ADAM17 signaling increases the number of voltage-dependent potassium channels in the membrane of the muscle cells. Having too many of these channels impairs the flow of blood through vessels in the brain.

Further experiments are needed to investigate whether correcting TIMP3-ADAM17 signaling could prevent strokes in people with inherited CADASIL. It also remains to be seen whether similar signaling mechanisms are at play in other small vessel diseases.

feature of the disease, suggesting that cerebrovascular dysfunction may have a key role in disease pathogenesis (*Chabriat et al., 2000*; *Pfefferkorn et al., 2001*; *Liem et al., 2009*).

Small vessels of the brain have unique functional properties that ensure that the brain, which has a limited capacity to store energy, maintains an adequate supply of blood-borne nutrients in the face of variations in blood pressure and changing neuronal energy demands. Cerebral arteries exist in a partially constricted state called 'myogenic tone', which reflects an intrinsic contractile response of arterial myocytes to increases in intravascular pressure. Thus, these arteries are positioned to dilate, and thereby increase local CBF, in response to elevated neuronal activity. This phenomenon, known as functional hyperemia, serves to satisfy enhanced glucose and oxygen demands of active neurons (*Iadecola and Nedergaard, 2007*). Impaired functional hyperemia and CBF autoregulation, attenuated CBF responses to topical application of vasodilators, and diminished myogenic responses of cerebral arteries and arterioles are early and prominent features of the well-established $TgNotch3^{R169C}$ CADASIL mouse model (*Joutel et al., 2010*; *Dabertrand et al., 2015*; *Capone et al., 2016*). The mechanisms underlying this cerebrovascular dysfunction are poorly understood.

Recently, we found that TIMP3 (tissue inhibitor of metalloproteinases-3) forms complexes with Notch3$^{ECD}$ and abnormally accumulates in the extracellular matrix of brain vessels of patients and mice with CADASIL (*Monet-Leprêtre et al., 2013*). Remarkably, genetic overexpression of TIMP3 recapitulates both CBF and myogenic-response deficits of the CADASIL model; conversely, genetic reduction of TIMP3 in CADASIL model mice restores normal function (*Capone et al., 2016*). TIMP family members are key regulators of the metalloproteinases that degrade the extracellular matrix. Within the TIMP family, TIMP3 has the broadest spectrum of substrates, which extends to members of the ADAM (a disintegrin and metalloproteinases) family. These metalloproteinases proteolytically release the extracellular domains of membrane-bound cytokines, cell adhesion molecules and growth factors, such as tumor necrosis factor-α and several ligands of the epidermal growth factor receptor (EGFR) family, including HB-EGF (heparin-binding EGF-like growth factor) (*Brew and Nagase, 2010*; *Khokha et al., 2013*; *Arpino et al., 2015*). As such, in addition to being a powerful regulator of extracellular matrix remodeling in various organs (*Arpino et al., 2015*), TIMP3 is a key player in inflammatory pathologies and autoimmune diseases through regulation of cell surface

proteins (*Brew and Nagase, 2010*; *Khokha et al., 2013*). However, how metalloproteinase inhibition might dynamically regulate arterial tone and CBF hemodynamics is unclear.

In another recent study, we established that upregulation of voltage-gated potassium ($K_V$) channels at the plasma membrane of arterial myocytes is responsible for the diminished myogenic responses of cerebral arteries and penetrating arterioles in the *TgNotch3$^{R169C}$* CADASIL model. Notably, an influence of the endothelium in myogenic tone deficit was ruled out (*Dabertrand et al., 2015*). $K_V$ channels play an important and dynamic role in opposing pressure-induced depolarization and vasoconstriction (*Longden et al., 2015*). Furthermore, we (*Dabertrand et al., 2015*) and our collaborators (*Koide et al., 2007*) have found that down-regulation of plasma membrane $K_V$ channels through activation of the EGFR pathway restores normal responses to pressure. Collectively, our results suggest a fundamental linkage between the activity of TIMP3 in the extracellular matrix of cerebral arterial myocytes and cerebral arterial tone.

Here, we find that the ADAM17/HB-EGF/EGFR (ErbB1/ErbB4) signaling axis is a key TIMP3-sensitive signaling module that regulates CBF responses and the myogenic tone of cerebral arteries. We further provide evidence that disruption of this TIMP3-sensitive pathway mediates cerebrovascular dysfunction in the *TgNotch3$^{R169C}$* CADASIL model and identify upregulated $K_V$ channel current density in cerebral arterial myocytes as a heretofore-unrecognized effector of this pathway. These insights into the relationship between TIMP3 activity and cerebral arterial tone may ultimately lead to therapeutic options in cerebral SVD.

## Results

### Exogenous TIMP3, but neither TIMP1 nor TIMP2, strongly attenuates CBF responses

To explore the role of TIMP3 in the regulation of CBF, we monitored CBF responses in wild-type mice equipped with an open cranial window over the somatosensory cortex, before and after the application of recombinant TIMP3 as well as TIMP1 or TIMP2 (*Figure 1A*; *Figure 1—source data 1*). We initially ensured that a recombinant protein applied over the cranial window could enter the brain. In the absence of robust anti-TIMP3 antibody suitable for immunohistochemistry and of an in situ assay to detect TIMP3 activity, we assessed brain penetration of Fluorescein isothiocyanate-labeled albumin (FITC-BSA, 66 kDa). After 30 min of continuous superfusion, fluorescence imaging of fixed vibratome slices showed that FITC-BSA entered the cortex along penetrating arteries beneath the cranial window (*Figure 1—figure supplement 1*), consistent with transport via the glymphatic system (*Iliff et al., 2012*). We found that TIMP3 (40 nM) did not affect resting CBF (*Figure 1B*), but did strongly reduce the increase in CBF evoked by whisker stimulation (*Figure 1C, D*; *Figure 1—source data 2,3*). Superfusion of 8 nM TIMP3 similarly attenuated functional hyperemia (*Figure 1—figure supplement 2*; *Figure 1—source data 2,3*). To rule out a possible effect of TIMP3 on neural activity, we recorded evoked neural activity during TIMP3 application (*Figure 1E*). We found that the amplitude of the somatosensory fields potentials produced by electrical stimulation of the whisker pad was unaltered by TIMP3 superfusion (*Figure 1F*).

The known TIMPs share 38–49% amino acid identity and inhibit most matrix metalloproteinases (MMPs). However, differences in substrate selectivity and inhibitory properties between different TIMPs have been described (*Khokha et al., 2013*; *Stetler-Stevenson, 2008*) (*Figure 1—source data 1*). This prompted us to assess the effects of other TIMPs on functional hyperemia. In sharp contrast to TIMP3, neither exogenous TIMP1 (50 nM) nor TIMP2 (50 nM) altered functional hyperemia (*Figure 1C,D*; *Figure 1—source data 2,3*).

We further assessed CBF responses to topical application of the endothelium-dependent and smooth muscle-dependent vasodilators, acetylcholine and adenosine, respectively, upon superfusion of TIMP3 (8 and 40 nM), TIMP1 (50 nM) or TIMP2 (50 nM). Again, the increases in CBF induced by acetylcholine or adenosine were profoundly attenuated by TIMP3 but were unaffected by TIMP2 or TIMP1, with the exception of a modest attenuation of the adenosine-induced increase in CBF by TIMP1 (*Figure 1G,H*; *Figure 1—figure supplement 2*; *Figure 1—source data 2,3*). Thus, these findings establish that elevation of TIMP3 is sufficient to induce CBF deficits in vivo and suggest that a TIMP3-specific target accounts for these deficits.

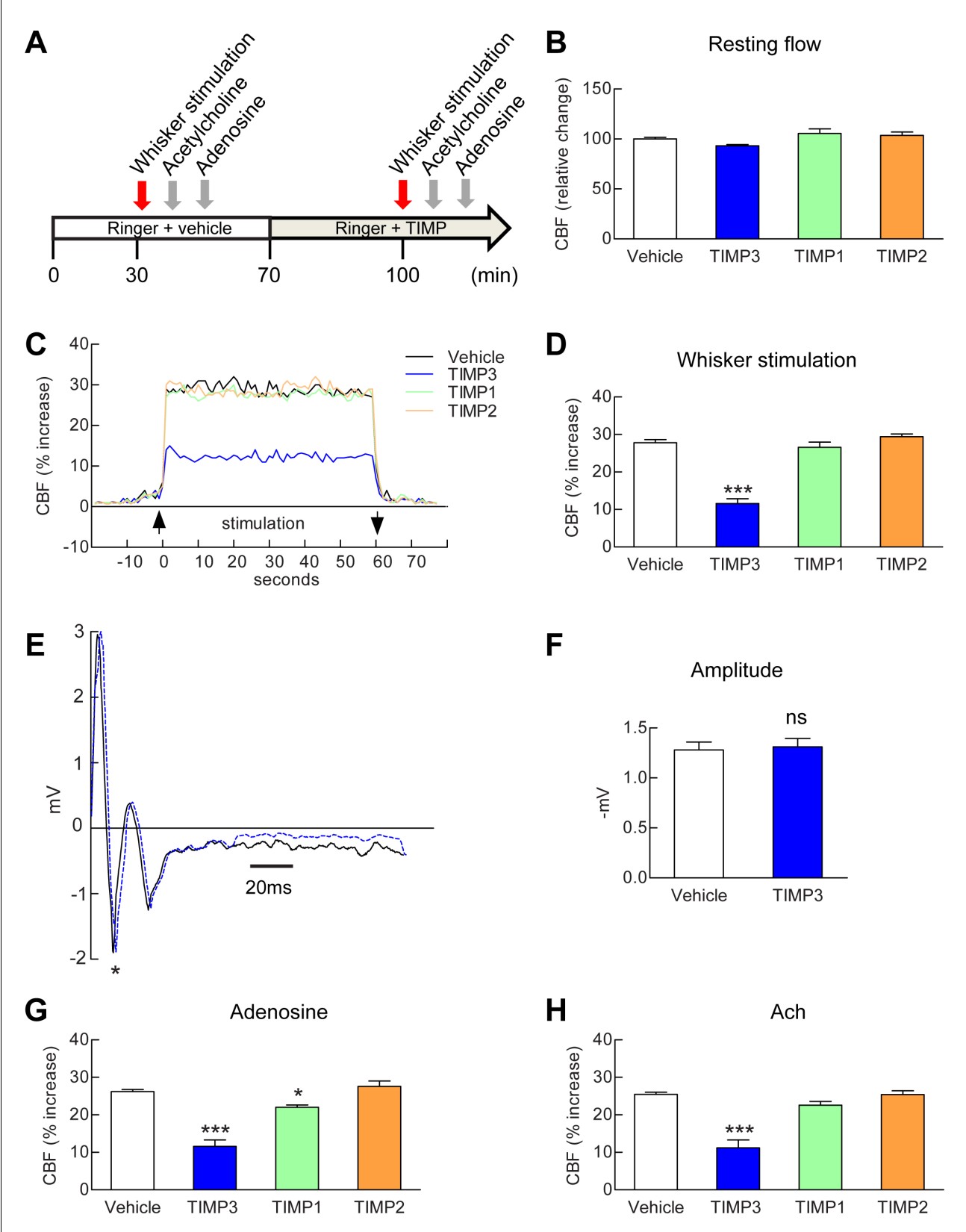

**Figure 1.** Exogenous TIMP3 specifically impairs cerebrovascular reactivity. (**A**) Schematic representation of the experimental protocol used to test the effects of recombinant TIMP1 (50 nM), TIMP2 (50 nM) or TIMP3 (40 nM) superfusion on the somatosensory cortex of 2-month-old wild-type mice. (**B–D**)
*Figure 1 continued on next page*

*Figure 1 continued*

Resting CBF (**B**) and CBF responses to whisker stimulation (**C, D**) were evaluated upon superfusion of vehicle or TIMP proteins. (**C**) Representative trace of CBF responses to whisker stimulation upon superfusion of vehicle or TIMP proteins (**C**). (**E**) Representative trace of the field potentials evoked by whisker stimulation upon vehicle or TIMP3 superfusion, showing typical sharp positive (P1)-negative (N1) waves followed by a slower positive-negative waveform occurring within 80 ms post stimulus (*Di and Barth, 1991*). (**F**) The amplitude of the negative wave (N1, asterisk in **E**) of the field potential was not affected by TIMP3 superfusion (p=0.79). (**G, H**) CBF responses to topical application of adenosine (**G**) or acetylcholine (**H**) upon superfusion of vehicle or TIMP proteins. Significance was determined by one-way ANOVA followed by Tukey's post-hoc test (**B, D, G, H**) or unpaired Student's t-test (**F**). (*p<0.05, ***p<0.001 compared to vehicle; n = 5 mice/groups). Error bars indicate SEM.

The following source data and figure supplements are available for figure 1:

**Source data 1.** Reagents used for *Figure 1*.

**Source data 2.** Main physiological variables of mice studied in *Figure 1*.

**Source data 3.** Numerical data that were used to generate the bar charts in *Figure 1*.

**Figure supplement 1.** Assessment of brain penetration of fluorescein isothiocyanate labelled serum albumin (FITC-BSA) superfused over the cranial window.

**Figure supplement 2.** Exogenous TIMP3 (8 nM) impairs cerebrovascular reactivity.

## ADAM17 is required for TIMP3-induced attenuation of CBF responses

Our efforts to identify the target of TIMP3 focused on ADAM17, which is uniquely inhibited by TIMP3 (*Xu et al., 2012*) and is expressed in brain arteries, as demonstrated by our immunoblot analyses (*Figure 2A,B*). If TIMP3 does indeed act through inhibition of ADAM17, its effects on CBF responses should be mimicked by pharmacological inhibition of ADAM17. Here, we used the hydroxamate-based GW413333X inhibitor, which specifically blocks both ADAM10 and ADAM17; the ADAM10 inhibitor GI254023X was used as a control (*Hundhausen et al., 2003*) (*Figure 2—source data 1*). We found that GW413333X (5 µM), but not GI254023X (5 and 20 µM), strongly attenuated the increase in CBF produced by whisker stimulation or topical application of acetylcholine or adenosine (*Figure 2C–E*; *Figure 2—figure supplement 1A*; *Figure 2—source data 2,3*). To further support a specific role for ADAM17 in these defects, we assessed CBF responses following reduction of ADAM17 levels using a genetic approach. Complete ablation of ADAM17 is lethal (*Peschon et al., 1998*). Therefore, we used hypomorphic mice with dramatically reduced expression of ADAM17 ($Adam17^{ex/ex}$) using the exon-induced translational stop strategy. These mice are viable, but develop eye, skin and heart defects as a consequence of impaired EGFR signaling (*Chalaris et al., 2010*). We found that genetic depletion of ADAM17 strongly attenuated CBF responses in a dose-dependent manner (*Figure 2F,G*; *Figure 2—figure supplement 1B*; *Figure 2—source data 2,3*). To confirm that the reduction in evoked CBF responses in these mice is caused by reduced ADAM17 expression, we examined whether an enzymatically active extracellular domain of ADAM17 (sADAM17) applied exogenously could prevent these CBF deficits. Topical neocortical application of sADAM17 (16 nM) (*Figure 2—source data 1*) did not affect cerebrovascular responses in wild-type mice, but did fully restore CBF responses in $Adam17^{ex/+}$ mice with half-reduced ADAM17 levels (*Figure 2H–J*; *Figure 2—figures supplements 1C,2A*; *Figure 2—source data 2,3*). Together, these results indicate that decreasing ADAM17 activity compromises CBF regulation.

To further confirm the direct connection between increased TIMP3 expression and reduced ADAM17 activity and CBF deficits, we tested whether exogenous sADAM17 is capable of preventing the CBF deficits produced by genetic overexpression of TIMP3. Superfusion with the enzymatically active extracellular domain of ADAM17 (16 nM) increased resting CBF in *TgBAC-TIMP3* mice towards the same absolute values as wild-type mice and improved all evoked cerebrovascular responses (*Figure 2K–M*; *Figure 2—figures supplements 1D,2B*; *Figure 2—source data 2,3*) suggesting that TIMP3 induces CBF deficits by decreasing ADAM17 activity.

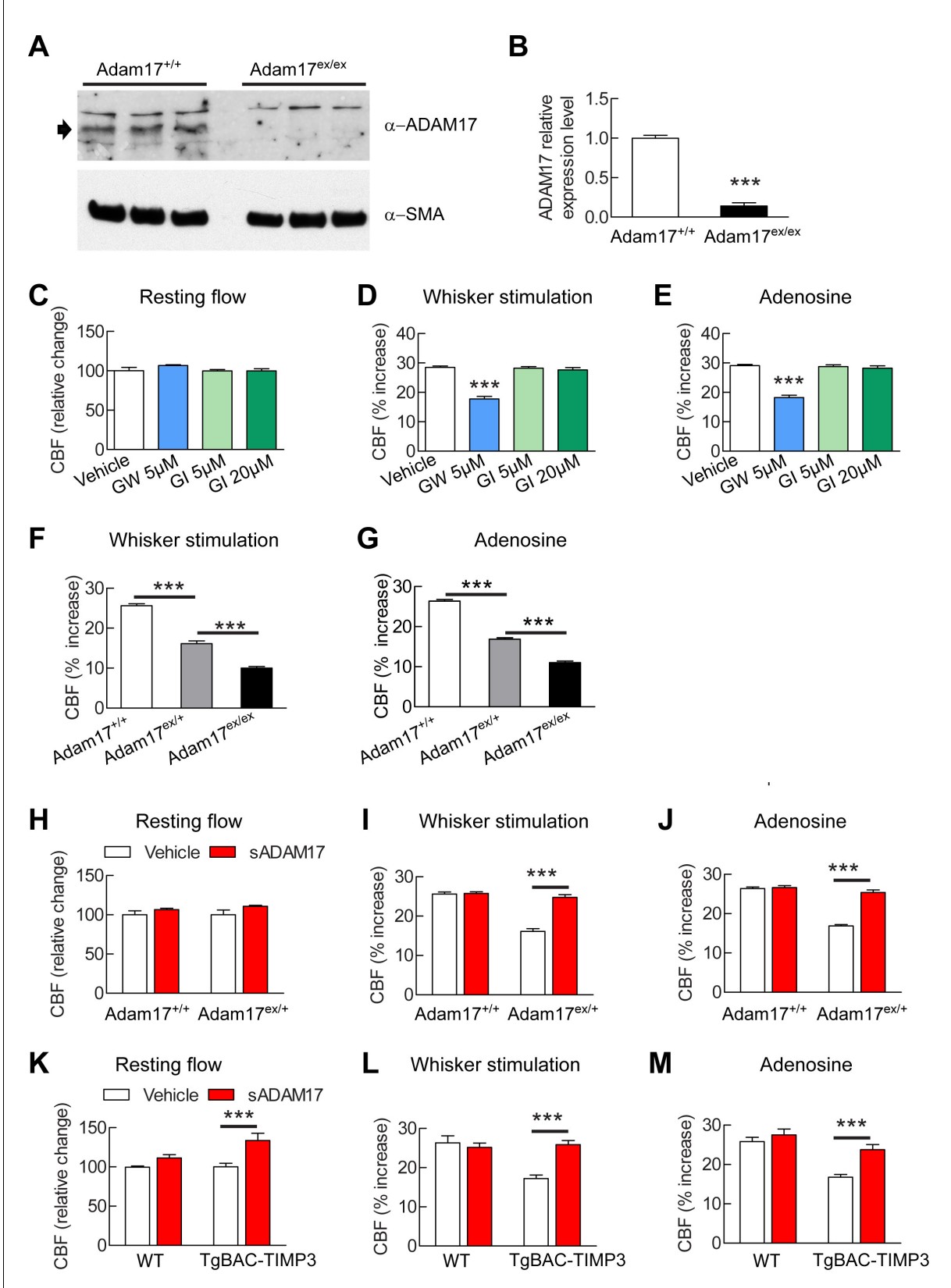

**Figure 2.** Cerebrovascular responses are impaired by pharmacological or genetic inhibition of ADAM17, and rescued by exogenous sADAM17. (**A**) Immunoblot of cerebral arteries dissected from *Adam17+/+* and *Adam17ex/ex* mice (n = 3 biological samples/genotype) incubated with anti-ADAM17 or
*Figure 2 continued on next page*

Figure 2 continued

anti-smooth muscle α-actin (α-SMA) antibody. (B) Quantification of relative protein level of ADAM17 in (A). (C–E) Resting CBF (C) and CBF responses to whisker stimulation (D) or topical application of adenosine (E) were evaluated upon superfusion of the dual ADAM10/ADAM17 inhibitor GW413333X (GW; 5 µM) or the ADAM10 inhibitor GI254023X (GI; 5 and 20 µM). ***p<0.05 compared with vehicle. (F, G) CBF responses to whisker stimulation (F) or topical application of adenosine (G) were strongly reduced in *Adam17*[ex/+] mice and further reduced in *Adam17*[ex/ex] mice compared with wild-type littermate controls. (H–J) Exogenous sADAM17 (16 nM) significantly ameliorated CBF responses to whisker stimulation (I) or topical application of adenosine (J) in *Adam17*[ex/+] mice, whereas ADAM17 had no effect on wild-type littermates. (K–M) Resting CBF and CBF responses were evaluated in *TgBAC-TIMP3* mice and non-transgenic littermates (WT) before and after superfusion of ADAM17. CBF responses to whisker stimulation (L) or topical application of adenosine (M) were strongly reduced in *TgBAC*-TIMP3 mice compared with those in WT mice, as previously reported (*Capone et al., 2016*), and were normalized by sADAM17 superfusion. Significance was determined by one-way ANOVA followed by Tukey's post-hoc test (B–G) and two-way repeated measure ANOVA followed by Bonferroni post-hoc test (H–M) (n = 5 mice/group). Error bars indicate SEM.

The following source data and figure supplements are available for figure 2:

**Source data 1.** Reagents used for *Figure 2*.
**Source data 2.** Main physiological variables of mice studied in *Figure 2*.
**Source data 3.** Numerical data that were used to generate the bar charts in *Figure 2*.
**Figure supplement 1.** CBF responses to acetylcholine are attenuated by pharmacological or genetic inhibition of ADAM17 but rescued upon superfusion of exogenous sADAM17.
**Figure supplement 2.** Absolute measurements of resting CBF in *Adam17*[ex/+] and *TgBAC-TIMP3* mice in the presence and absence of sADAM17.

## HB-EGF and EGFR operate downstream of ADAM17 to regulate CBF responses

To elucidate the molecular factors that operate downstream of ADAM17 in the context of cerebro-vascular regulation, we examined the role of the EGFR signaling pathway. This pathway consists of four related receptor tyrosine kinases of the ErbB family—ErbB1/EGFR (Her1), ErbB2/Neu (Her2), ErbB3 (Her3) and ErbB4 (Her4)—which are regulated by 11 different ligands, all of which are produced as membrane-bound precursor proteins and cleaved by cell surface proteases to yield the active soluble species; ADAM17 is the critical sheddase of at least six of these ligands (*Sahin et al., 2004*; *Roskoski, 2014*) (*Figure 3A,B*). A critical role for ADAM17 in EGFR signaling is supported by the observation that mice deficient for ADAM17 (*Peschon et al., 1998*; *Chalaris et al., 2010*) resemble mice lacking EGFR, exhibiting perinatal lethality, generalized epithelial defects, and defective cardiac valves (*Miettinen et al., 1995*; *Sibilia and Wagner, 1995*; *Threadgill et al., 1995*). To investigate the role of the EGFR pathway, we recorded CBF responses evoked by whisker stimulation or vasodilators before and after topical application of blockers of this pathway (*Figure 3—source data 1*).

Based on the decrease in evoked CBF responses to elevation of TIMP3 or reduction of ADAM17, we predicted that inhibition of ErbB pathway would have a similar effect. Indeed, we found that neo-cortical application of the selective ErbB1/EGFR and ErbB4 inhibitor, tyrphostin AG1478 (10 and 20 µM), strongly attenuated the evoked CBF responses, but did not impair resting CBF. In contrast, CBF responses were unaffected by the selective ErbB2 inhibitor, tyrphostin AG825, at both 50 and 200 µM (*Figure 3C–E*; *Figure 3—source data 2, 3*). Of note, ErbB3 lacks kinase activity (*Roskoski 2014*). We then tested the effect of soluble recombinant decoy ErbB receptors, known as chimeric ErbB receptor traps, which comprise the truncated extracellular domain of the ErbB receptor fused with the constant region (Fc) of human immunoglobulin (*Stratman et al., 2010*). Evoked CBF responses were attenuated by superfusion of either ErbB1/EGFR (67 nM) or ErbB4 (71 nM) receptor traps, which block the function of ErbB1 and ErbB4 ligands, respectively, but not by superfusion of the ErbB3 (71 nM) receptor trap or by control IgG1 Fc (286 nM) and IgG2 Fc (286 nM) fragments (*Figure 3F–H*; *Figure 3—figure supplement 1A*; *Figure 3—source data 2, 3*). Notably, the effects of ErbB1 and ErbB4 receptor traps on evoked CBF responses were not additive (*Figure 3G–H*; *Figure 3—source data 2, 3*), even though neither ErbB1 nor ErbB4 receptor traps achieved maximum inhibition. ErbB2 has no known ligand (*Roskoski, 2014*); thus, these data are consistent with a role

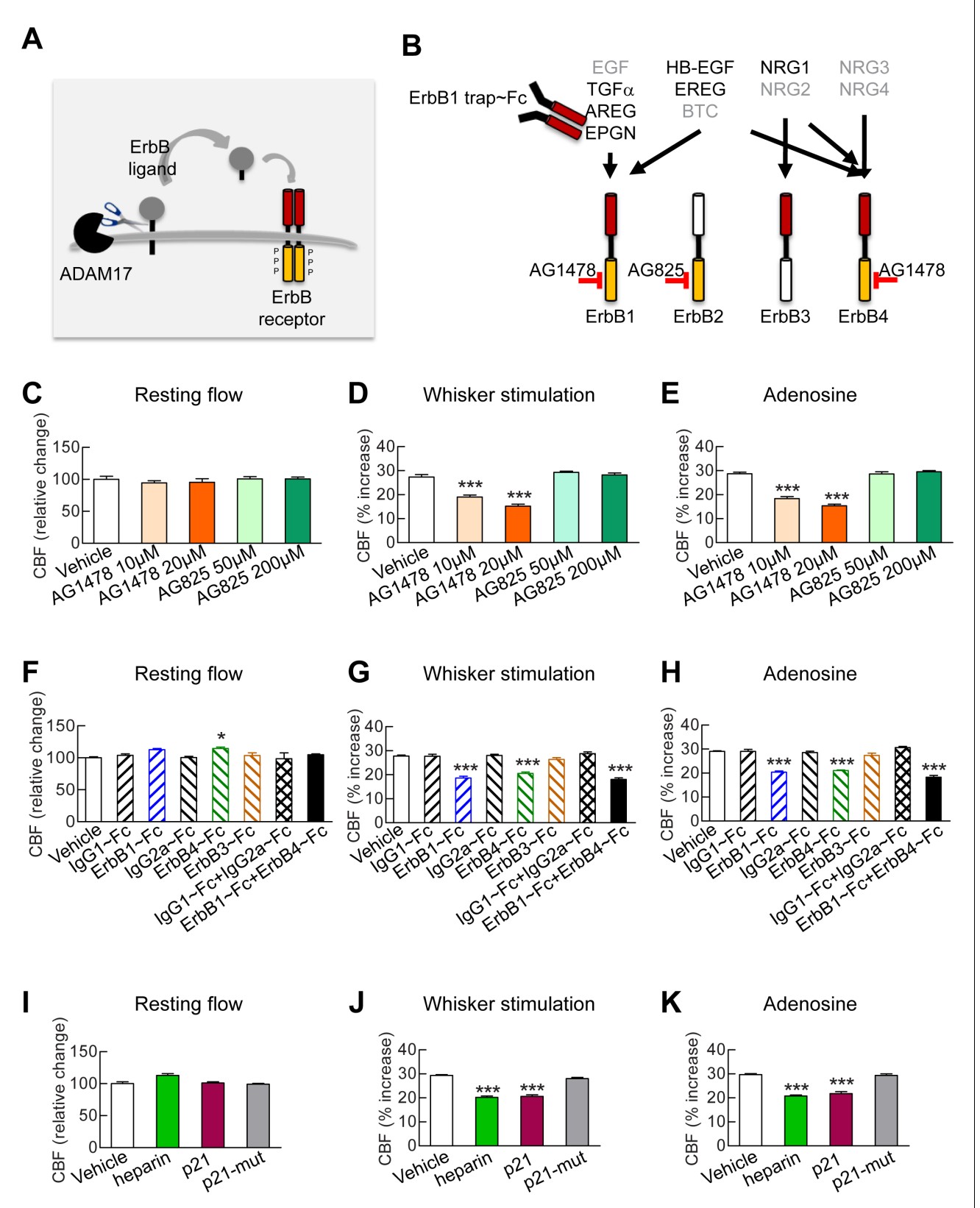

**Figure 3.** Full CBF responses require ErbB1/ErbB4 and HB-EGF. (**A**, **B**) Schematic representation of the ErbB signaling pathway. Ligands are all produced as membrane-bound precursor proteins that are cleaved by cell-surface sheddases to yield the active growth factor species. Binding of the

*Figure 3 continued on next page*

*Figure 3 continued*

soluble form of the ligand induces ErbB receptor homodimerization or heterodimerization, converting the receptor to an active dimeric conformation (A). Ligands are grouped in four rows according to their receptor specificity (top; arrows); the six ligands for which ectodomain shedding is primarily mediated by ADAM17 appear in black characters, and the remaining five are in grey characters (B). (C–K) Resting CBF (C, F, I) and CBF responses to whisker stimulation (D, G, J) or topical application of adenosine (E, H, K) were evaluated before and after superfusion of various inhibitors of the ErbB signaling pathway, including the ErbB1/ErbB4 inhibitor AG1478 (10 and 20 µM); the ErbB2 inhibitor AG825 (50 and 200 µM) (C–E), the soluble ErbB receptor traps (ErbB1-Fc, 66.7 nM; ErbB3-Fc, 71.4 nM; ErbB4-Fc, 71.4 nM) and the respective control IgG1-Fc and IgG2-Fc fragments (286 nM) (F–H), heparin and the synthetic peptide p21 (12 µM) and the control inactive peptide p21-mut (12 µM) (I–K). None of these compounds affected resting CBF, except ErbB4-Fc, which produced a slight increase. (C–K) Significance was determined by one-way ANOVA followed by Tukey's post-hoc test (*$p<0.05$, **$p<0.01$, ***$p<0.001$ compared to vehicle; n = 5/group). Error bars indicate SEM.

The following source data and figure supplement are available for figure 3:

**Source data 1.** Reagents used for *Figure 3*.
**Source data 2.** Main physiological variables of mice studied in *Figure 3*.
**Source data 3.** Numerical data that were used to generate the bar charts in *Figure 3*.
**Figure supplement 1.** Blockade of ErbB1/ErbB4 or HB-EGF impairs CBF responses to acetylcholine.

for ErbB1/EGFR or ErbB4 activation in CBF responses, and suggest the involvement of bispecific ligands with dual-specificity toward ErbB1 and ErbB4.

Next, we sought to pinpoint which ErbB ligand that requires *ADAM17* cleavage for activation is involved in CBF regulation. Heparin-binding EGF-like growth factor (HB-EGF) is one of three ligands that can bind to both ErbB1 and ErbB4 and is expressed in the vasculature (*Zhang et al., 2014*). ADAM17 is the major sheddase of HB-EGF (*Sahin et al., 2004*), and ADAM17-mediated shedding of proHB-EGF largely regulates soluble, mature HB-EGF binding to and activating ErbB receptors (*Yamazaki et al., 2003*). Moreover, mice lacking HB-EGF have reduced postnatal viability with defective cardiac valvulogenesis, similar to mice lacking ADAM17 (*Jackson et al., 2003*), prompting us to study the role of HB-EGF in cerebrovascular regulation. To do this, we examined the impact of HB-EGF inhibition on CBF responses. Unlike all other EGF ligands apart from amphiregulin, HB-EGF has a heparin-binding domain, and interactions through this domain with cell surface-associated heparan sulfate proteoglycans (HSPGs) are necessary for binding and activation of ErbB receptors (*Higashiyama et al., 1993*). We found that superfusion of heparin (40 ui/mL), which competitively inhibits binding of HB-EGF to cell surface HSPGs (*Higashiyama et al., 1993*), impaired evoked CBF responses without affecting resting CBF (*Figure 3I–K*; *Figure 3—figure supplement 1B*; *Figure 3—source data 2*, *3*). To further support a role for HB-EGF in evoked CBF responses, we examined the effects of the synthetic peptide p21, which corresponds to the heparin-binding domain of murine HB-EGF and similarly inhibits binding of HB-EGF to cell surface HSPGs (*Higashiyama et al., 1993*). We found that superfusion of p21 (12 µM) similarly impaired evoked CBF responses without affecting resting CBF (*Figure 3I–K*; *Figure 3—figure supplement 1B*; *Figure 3—source data 2*, *3*); in contrast, a mutated inactive p21 peptide (p21-mut; 12 µM) had no effect on evoked or resting CBF.

To assess the connection between HB-EGF and ADAM17 in the context of cerebrovascular regulation, we tested the ability of a soluble form of HB-EGF (sHB-EGF) to counteract cerebrovascular dysfunction induced by ADAM17 inhibition or depletion (*Figure 4—source data 1*). TIMP3 or the ADAM10/17 inhibitor, GW413333X, was topically applied over the neocortex and CBF responses were measured before and after superfusion with sHB-EGF. We found that sHB-EGF (20 nM) prevented TIMP3 and GW-induced cerebrovascular deficits (*Figure 4D–I*; *Figure 4—figure supplement 1A,B*; *Figure 4—source data 2,3*). Also, sHB-EGF significantly improved evoked CBF responses in *Adam17*$^{ex/ex}$ mice (*Figure 4—figure supplement 2*). Notably, sHB-EGF could not prevent CBF deficits induced by pharmacological blockage of ErbB1/EGFR and ErbB4 (*Figure 4A–C*; *Figure 4—source data 1–3*). These findings, collectively, suggest that ADAM17/HB-EGF/(ErbB1/ErbB4) is a key TIMP3-sensitive signaling pathway for cerebrovascular regulation.

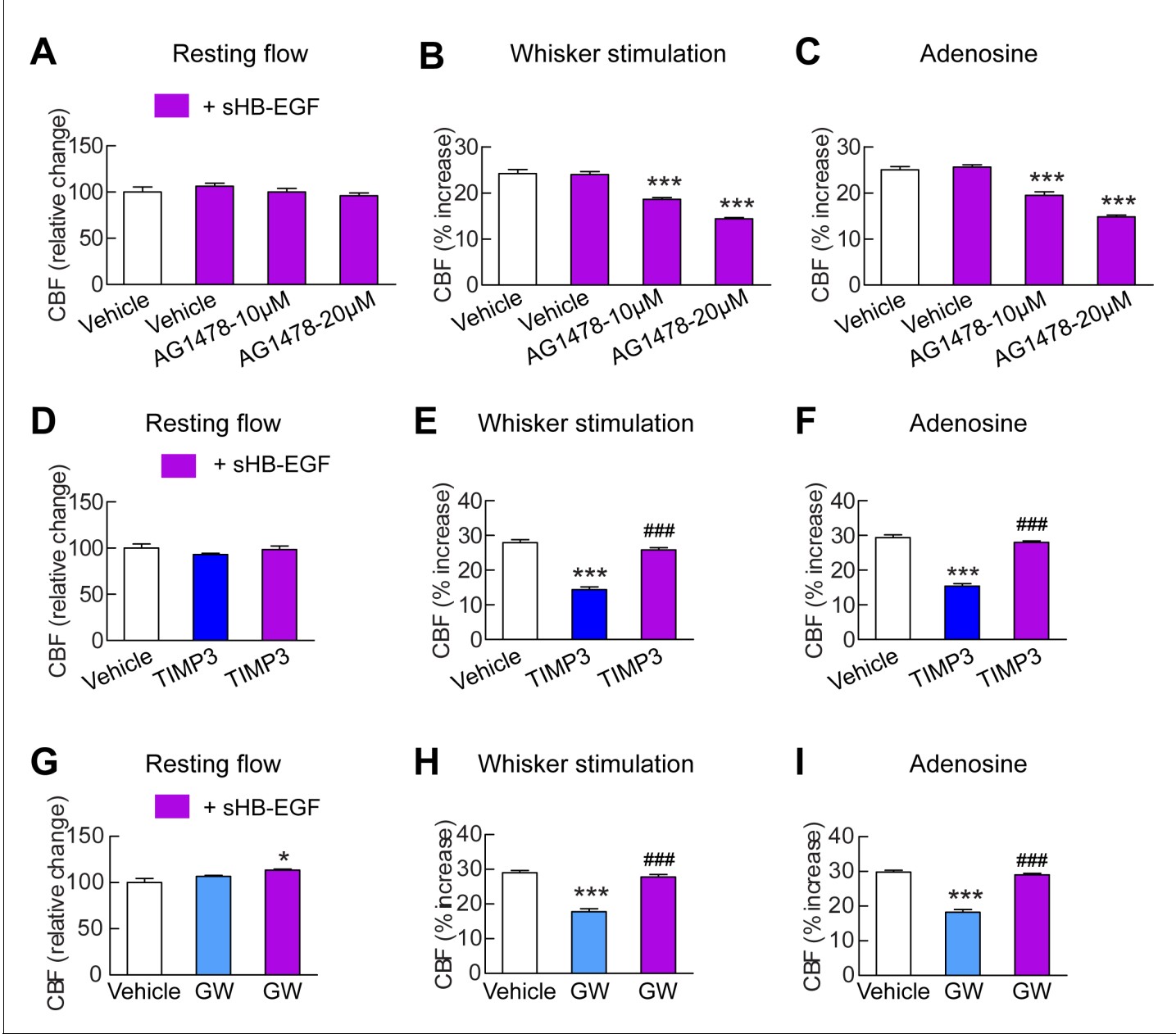

**Figure 4.** sHB-EGF overcomes CBF deficits induced by ADAM17 inhibition. Effects of sHB-EGF (20 nM) on resting CBF (**A**, **D**, **G**) and whisker stimulation (**B**, **E**, **H**)- and adenosine (**C**, **F**, **I**)-induced CBF responses were assessed in the presence and absence of the ErbB1/ErbB4 inhibitor AG1478 (10 and 20 µM) (**A**–**C**), TIMP3 (40 nM) (**D**–**F**) or the ADAM10/ADAM17 inhibitor GW413333X (GW; 5 µM) (**G**–**I**) using a cranial window model. Significance was determined by repeated measure ANOVA followed by Tukey's post-hoc test (*$p<0.05$, ***$p<0.001$ compared to vehicle; ###$p<0.001$ TIMP3+sHB-EGF versus TIMP3 and GW+sHB-EGF versus GW; n = 5/group). Error bars indicate SEM.

The following source data and figure supplements are available for figure 4:

**Source data 1.** Reagents used for *Figure 4*.

**Source data 2.** Main physiological variables of mice studied in *Figure 4*.

**Source data 3.** Numerical data that were used to generate the bar charts in *Figure 4*.

**Figure supplement 1.** Acetylcholine-induced CBF responses impaired by ADAM17 inhibition are ameliorated by exogenous sHB-EGF.

*Figure 4 continued*

**Figure supplement 2.** CBF deficits induced by ADAM17 deficiency are improved by sHB-EGF.

## The ADAM17/HB-EGF/(ErbB1/ErbB4) signaling module regulates pressure-induced myogenic tone in brain arteries

Our data support the concept that the ADAM17/HB-EGF/(ErbB1/ErB4) axis regulates CBF responses, and any genetic or pharmacological maneuver that inhibits this pathway impairs CBF responses to diverse stimuli, including topical application of vasodilators and neural activity (*Figures 1–3*). Notably, responses cannot be enhanced by stimulation of this pathway, implying that this pathway is maximally activated in a physiological in vivo setting. Given that myogenic tone sets the resting arterial diameter from which other stimuli can induce vasoconstriction or vasodilation, we hypothesized that a reduction in the myogenic tone of cerebral arteries could represent a common mechanism underlying these CBF deficits. To test this hypothesis, we assessed the effects of inhibitors of this pathway on pressure-induced constriction of brain arteries (*Figure 5—source data 1*).

We found that pre-incubation of arterial segments with TIMP3 (8 nM) strongly attenuated myogenic tone at pressures of 40 mmHg and above compared to arteries incubated with vehicle, whereas recombinant TIMP2 (10 nM) had no effect (*Figure 5A–C*; *Figure 5—source data 2*). Notably, attenuation of myogenic responses by TIMP3 was even more pronounced in intracerebral penetrating arterioles (*Figure 5—figure supplement 1*). Likewise, myogenic constriction to pressure was strongly attenuated by the dual ADAM10/ADAM17 inhibitor GW413333X (1 µM), but not by the ADAM10 inhibitor GI254023X (1 µM). Also, myogenic tone was reduced in heterozygous ADAM17 hypomorphic mice (*Adam17^{ex/+}*) compared to wild-type littermates (*Adam17^{+/+}*) but restored by pre-incubating arterial segments of *Adam17^{ex/+}* mice with sADAM17 (3.2 nM) (*Figure 5D,I*; *Figure 5—figure supplement 2*; *Figure 5—source data 2*). Moreover, pre-incubation of arteries with the ErbB1/ErbB4 inhibitor AG1478 (2 µM) or the HB-EGF inhibitor p21 peptide (2.4 µM), but not with the p21-mut peptide (2.4 µM), strongly attenuated myogenic responses (*Figure 5E,F*; *Figure 5—source data 2*). Thus, these data indicate that myogenic tone is increased by tonic activity of the ADAM17/HB-EGF/(ErbB1/ErbB4) pathway.

To provide further support for the concept that ADAM17 and HB-EGF function as part of a signaling module to enhance the myogenic tone of cerebral arteries, we tested the ability of exogenous sHB-EGF to counteract the effects of ADAM17 inhibition (*Figure 5—source data 1*). Pressurized arteries were pre-incubated with recombinant TIMP3 (4 nM) or the ADAM10/ADAM17 inhibitor GW413333 in the presence of sHB-EGF (3 nM) or vehicle. We found that co-incubation of arterial segments with sHB-EGF significantly ameliorated the TIMP3-induced reduction in arterial tone (*Figure 5G*; *Figure 5—source data 2*). Likewise, co-incubation of arterial segments with sHB-EGF overcame the reduction in arterial tone caused by GW413333-mediated inhibition of ADAM17 (*Figure 5H*; *Figure 5—source data 2*). Moreover, sHB-EGF significantly increased myogenic tone in arteries from heterozygous ADAM17-hypomorphic mice (*Adam17^{ex/+}*), restoring a near-normal myogenic phenotype (*Figure 5I*; *Figure 5—source data 2*). Collectively, these data support the concept that the TIMP3-sensitive pathway, ADAM17/HB-EGF/(ErbB1/ErbB4), increases myogenic constriction in brain arteries.

## Exogenous sADAM17 and exogenous sHB-EGF rescue CBF and myogenic-response deficits in the *TgNotch3^{R169C}* CADASIL model

Our findings above predict that excess TIMP3 impairs arterial tone and CBF responses in CADASIL by suppressing the ADAM17/HB-EGF/(ErbB1/ErbB4) pathway. To test this, we examined whether recombinant sADAM17 and sHB-EGF could restore normal pressure-induced myogenic constriction of brain arteries and normal cerebrovascular responses in *TgNotch3^{R169C}* CADASIL mice. We found that preincubation of arterial segments with the enzymatically active extracellular domain of ADAM17 (3.2 nM) increased myogenic tone in arteries from *TgNotch3^{R169C}* CADASIL mice, whereas sADAM17 had no detectable effect on arterial segments from wild-type mice at this concentration (*Figure 6A*). We further found that sADAM17 (16 nM), locally applied on the necortex of *TgNotch3^{R169C}* CADASIL mice, significantly improved resting CBF and rescued the impaired

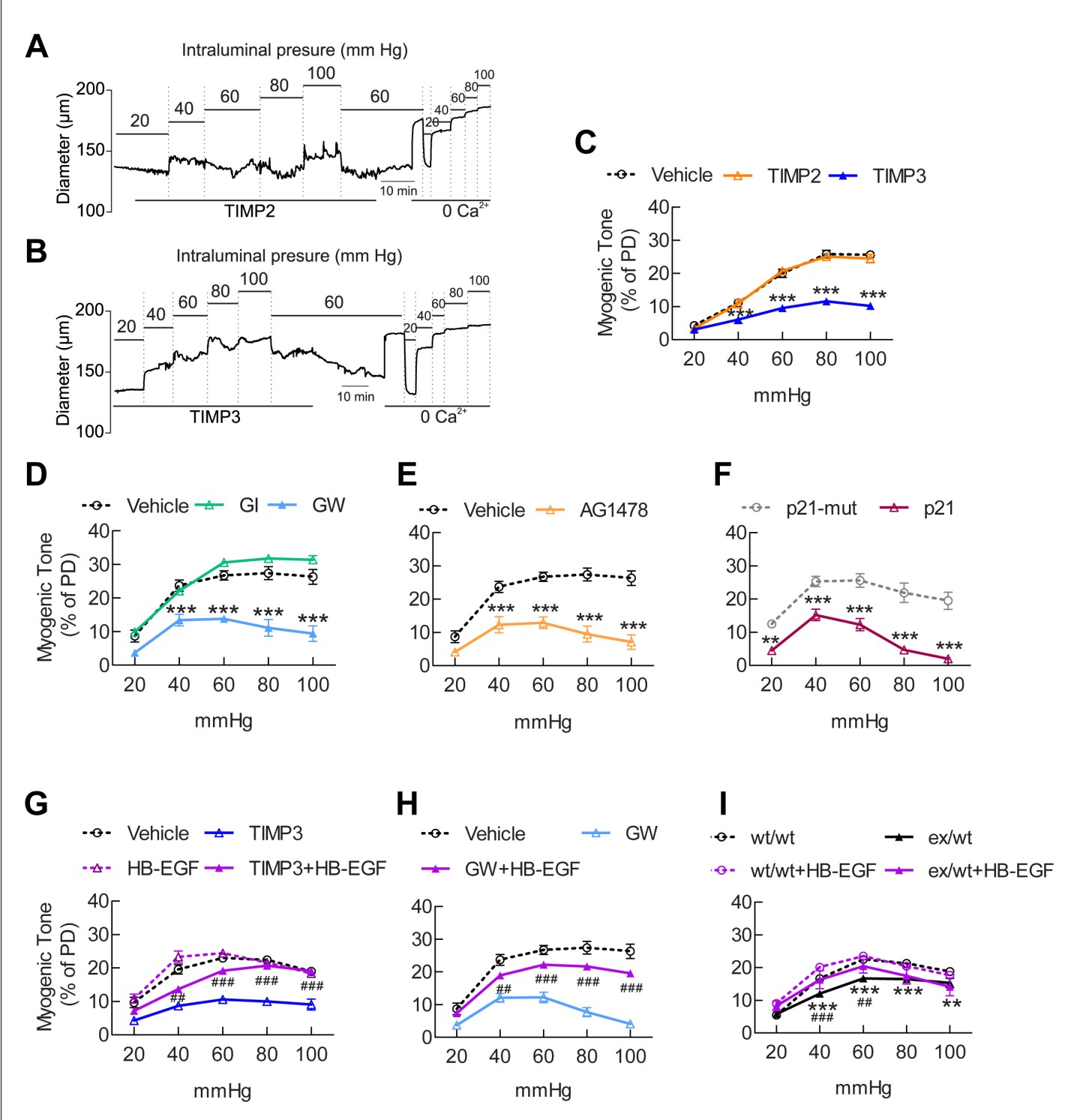

**Figure 5.** The ADAM17/HB-EGF/(ErbB1/ErbB4) signaling module is involved in regulating the myogenic tone of cerebral arteries. (A–C) Effects of TIMP proteins on the myogenic responses of posterior cerebral arteries to increasing intraluminal pressure. (A, B) Representative internal diameter recordings in the presence of TIMP2 (10 nM) (A) or TIMP3 (8 nM) (B). (C) Summary data of results in (A) and (B). (D–F) Myogenic tone of posterior cerebral arteries, tested in the presence and absence of the dual ADAM10/ADAM17 inhibitor GW413333X (GW; 1 μM), the ADAM10 inhibitor GI254023X (GI; 1 μM) (D), the ErbB1/ErbB4 inhibitor AG1478 (2 μM) (E), the p21 peptide (2.4 μM), and the mutated inactive peptide p21-mut (2.4 μM) (F). (C–F) **p<0.01, ***p<0.001 versus vehicle. (G, H) Effects of TIMP3 (8 nM) (g) or GW (1 μM) (H) on the myogenic tone of posterior cerebral arteries were tested in the presence of soluble HB-EGF (3 nM) or vehicle. ##p<0.01, ###p<0.001, TIMP3+HB-EGF versus TIMP3 and GW+HB-EGF versus GW. (I) Myogenic tone of

*Figure 5 continued on next page*

Figure 5 continued

posterior cerebral arteries was tested in heterozygous *Adam17*$^{ex/+}$ (ex/wt) and *Adam17*$^{+/+}$(wt/wt) mice in the presence and absence of soluble HB-EGF (3 nM). **p<0.01, ***p<0.001 *Adam17*$^{ex/+}$ versus *Adam17*$^{+/+}$ ; ##p<0.01, ###p<0.001, *Adam17*$^{ex/+}$/HB-EGF versus *Adam17*$^{ex/+}$). Significance was determined by two-way repeated measures ANOVA followed by Bonferroni post-hoc test (n = 6–8 arteries/group). Error bars indicate SEM.

The following source data and figure supplements are available for figure 5:

**Source data 1.** Reagents used for *Figure 5*.
**Source data 2.** Numerical data that were used to generate the graphs in *Figure 5*.
**Figure supplement 1.** TIMP3 strongly impairs myogenic tone of parenchymal arterioles.
**Figure supplement 2.** sADAM17 ameliorates arterial tone in *Adam17*$^{ex/+}$ mice.

reactivity of brain vessels to whisker stimulation and vasodilators (*Figure 6B–D*; *Figure 6—figure supplement 1*; *Figure 6—source data 1,2*). We previously reported that sHB-EGF restores myogenic responses in parenchymal arteries from *TgNotch3*$^{R169C}$ CADASIL mice (*Dabertrand et al., 2015*). Here, we extend these observations, showing that exogenous sHB-EGF (20 nM) restored evoked CBF responses in *TgNotch3*$^{R169C}$ mice (*Figure 6E–G*; *Figure 6—source data 1,2*). Collectively, these findings support the concept that the diminished myogenic tone and CBF deficits in CADASIL are caused by TIMP3-mediated suppression of the ADAM17/HB-EGF/(ErbB1/ErbB4) pathway.

## Excess TIMP3 drives upregulation of $K_V$ currents in cerebral arterial myocytes from *TgNotch3*$^{R169C}$ mice

Our prior work established that upregulation of $K_V$ channels in the plasma membrane of cerebral arterial myocytes is responsible for the diminished myogenic response of cerebral arteries in the *TgNotch3*$^{R169C}$ CADASIL model. Importantly, application of sHB-EGF was found to normalize $K_V$ current density and restore myogenic responses in cerebral arteries from *TgNotch3*$^{R169C}$ mice (*Dabertrand et al., 2015*). In light of this and the above, we investigated the involvement of TIMP3 and ADAM17 in this upregulation of $K_V$ current density.

We first asked whether reducing TIMP3 expression in the *TgNotch3*$^{R169C}$ mice would decrease the number of functional $K_V$ channels. To this end, we measured $K_V$ currents in freshly isolated myocytes from cerebral arteries of *TgNotch3*$^{R169C}$ mice with normal expression of TIMP3 (*TgNotch3*$^{R169C}$;*Timp3*$^{+/+}$), which have reduced myogenic tone, and in freshly isolated myocytes from cerebral arteries of *TgNotch3*$^{R169C}$ mice with reduced expression of TIMP3 (*TgNotch3*$^{R169C}$; *Timp3*$^{+/-}$), in which myogenic responses are restored (*Dabertrand et al., 2015*). Currents were recorded in response to 10-mV voltage steps from −70 mV to +60 mV. We found that $K_V$ current density was significantly lower in myocytes from *TgNotch3*$^{R169C}$;*Timp3*$^{+/-}$ mice than in myocytes from *TgNotch3*$^{R169C}$;*Timp3*$^{+/+}$ at all voltage steps above +10 mV (*Figure 7A–C*; *Figure 7—source data 2*). Conversely, incubation of wild-type arterial myocytes with recombinant TIMP3 (8 nM) resulted in a significant increase in $K_V$ current density compared with myocytes incubated with vehicle (*Figure 7—figure supplement 1A,B*; *Figure 7—source data 2*). Remarkably, half-maximal activation voltage ($V_{0.5}$) and slope (k), determined by fitting normalized peak tail currents to the Boltzmann equation, were statistically indistinguishable among arterial myocytes from the different groups analyzed. Likewise, activation ($\tau_{act}$) and deactivation ($\tau_{deact}$) time constants determined from exponential fits of individual voltage-evoked current traces and current decay, respectively, were comparable among the different groups. These current kinetics, attributable to $K_V1.5$ channels, are consistent with our previous report (*Dabertrand et al., 2015*). These results indicate that the TIMP3 pathway regulates the number of channels, and not channel properties (*Figure 7—figure supplement 2*; *Figure 7—source data 1,2*). Using the Goldman–Hodgkin–Katz constant field equation and a single-channel conductance of 15 pS (*Aiello et al., 1998*), we estimated the average number of functional $K_V$ channels per myocyte. This analysis showed that exogenously applied TIMP3 increased the number of $K_V$ channels in arteries from wild-type mice by ~25% (from 3120 to 3920 per myocyte).

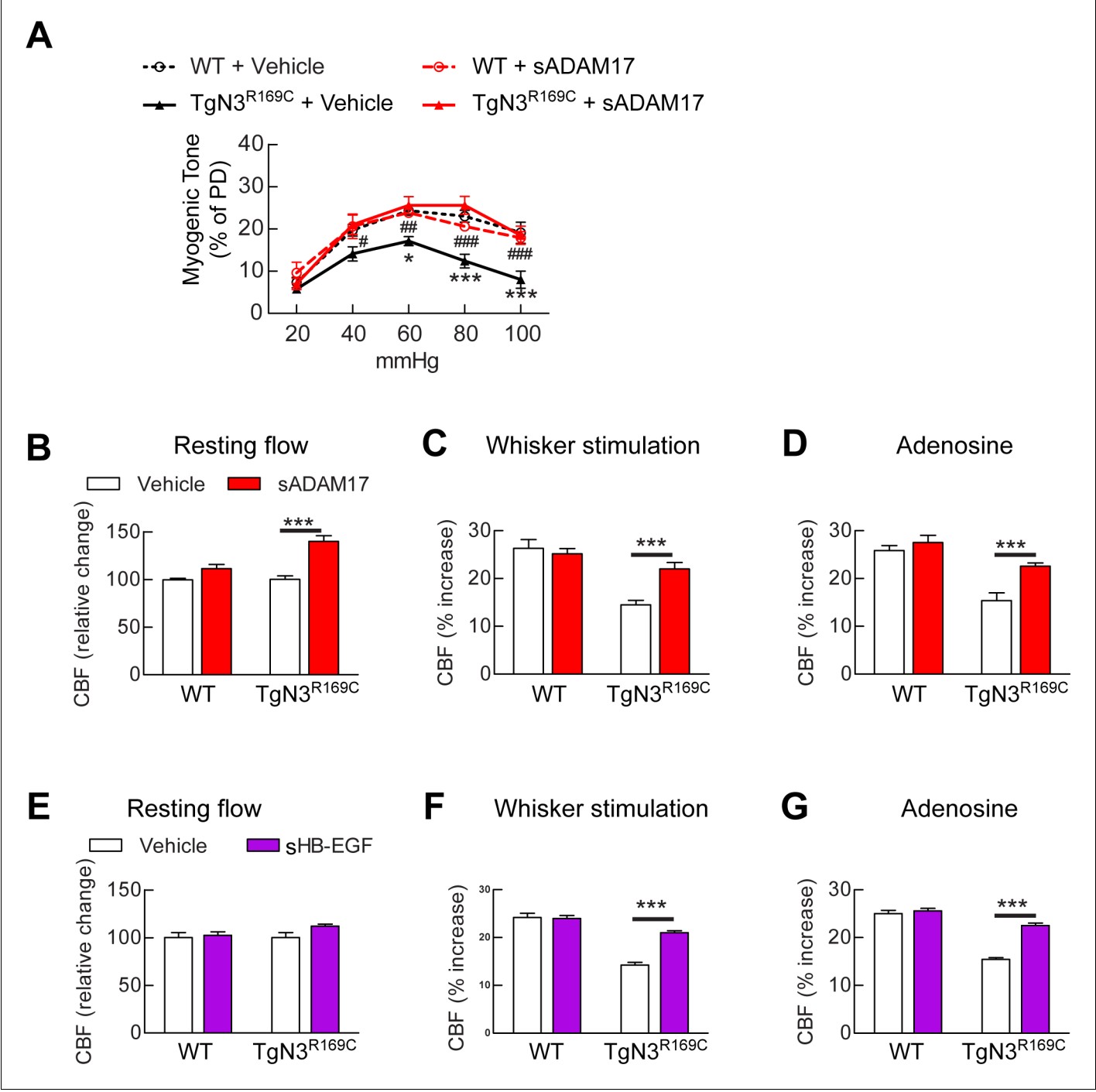

**Figure 6.** Exogenous sADAM17 and sHB-EGF ameliorate CBF deficits and arterial tone in *TgNotch3*[R169C] mice. (**A**) Myogenic tone of posterior cerebral arteries from *TgNotch3*[R169C] mice (TgN3[R169C]) and non-transgenic littermates (WT) was tested in the presence of soluble ADAM17 or vehicle. *$p<0.05$, **$p<0.01$, ***$p<0.001$ versus WT+vehicle; #$p<0.05$, ##$p<0.01$, ###$p<0.001$ TgN3[R169C]+vehicle versus TgN3[R169C]+sADAM17 (n = 5–7 arteries/ group; 1 artery/mouse). (**B–D**) Resting CBF (**B**) and CBF responses to whisker stimulation (**C**) or adenosine (**D**) were tested in TgN3[R169C] and WT mice, before and after superfusion of soluble ADAM17. (**E–G**) Effects of soluble HB-EGF tested in a second batch of TgN3[R169C] and WT mice. Significance was determined by two-way repeated measures ANOVA followed by Bonferroni post-hoc test (n = 5–6 mice/group). Error bars indicate SEM.

The following source data and figure supplement are available for figure 6:

**Source data 1.** Main physiological variables of mice studied in *Figure 6*.

*Figure 6 continued on next page*

*Figure 6 continued*

**Source data 2.** Numerical data that were used to generate the graphs and bar charts in *Figure 6*.

**Figure supplement 1.** Resting CBF and acetylcholine-induced CBF responses impaired by the R169C Notch3 mutation are ameliorated by exogenous sADAM17.

A similar increase in $K_V$ channel number was observed in the TIMP3-overexpressing $TgNotch3^{R169C}$; $Timp3^{+/+}$ genetic model, where channel density (4840/myocyte) was ~38% greater than that in $TgNotch3^{R169C}$;$Timp3^{+/-}$ mice (3510/myocyte).

Our model predicts that exogenous sADAM17 should counteract the increase in TIMP3 in cerebral arteries of $TgNotch3^{R169C}$ mice by decreasing $K_V$ channel density. Consistent with the ability of exogenous sADAM17 to restore normal myogenic responses in $TgNotch3^{R169C}$ mice, we found that application of enzymatically active, sADAM17 (3.2 nM) significantly reduced $K_V$ current density in $TgNotch3^{R169C}$ cerebral myocytes, decreasing the density of Kv channels by ~22% (from 4840 to 3760 channels per myocyte) (*Figure 7D,E*; *Figure 7—source data 2*). Thus, these new findings, taken together with our previous studies, indicate that excess TIMP3 in the $TgNotch3^{R169C}$CADASIL-model drives increased $K_V$ channel density and diminished myogenic responses by reducing ADAM17 activity and subsequently reducing the release of sHB-EGF.

## Discussion

Although SVD of the brain is a heterogeneous group of disorders with different ultimate causes acting through specific pathways, the recently emerging view is that perturbations of proteins constituting or associated with the extracellular matrix of cerebral vessels could be a convergent pathway driving the functional and structural alterations of small brain vessels (*Joutel et al., 2016*). Previously, we demonstrated that elevated TIMP3, a protein tightly bound to the extracellular matrix of brain arteries, contributes to cerebrovascular dysfunction in CADASIL, a genetic paradigm of small vessel disease of the brain (*Monet-Leprêtre et al., 2013*; *Capone et al., 2016*). In the present study, we establish the novel concept that a TIMP3-sensitive pathway is constitutively engaged in the regulation of cerebral hemodynamics, and we unravel the mechanism by which excess TIMP3 in brain vessels compromises cerebrovascular regulation in a clinically relevant model of CADASIL.

By combining genetic and pharmacological approaches with in vivo analyses of CBF regulation and ex vivo measurements of myogenic responses of brain arteries in physiological settings, we found that ADAM17/HB-EGF/(ErbB1/ErbB4) is a key TIMP3-sensitive signaling module essential for maintaining robust CBF responses to evoked neural activity or topically applied vasodilators as well as for myogenic responses of brain arteries. Next, using the $TgNotch3^{R169C}$ model, we provided pharmacological evidence that, in the setting of CADASIL, attenuated ADAM17 and HB-EGF-dependent activation of ErbB1/ErbB4 underlies deficits in evoked CBF responses and cerebral arterial tone. Further, by using patch clamp electrophysiology in combination with genetic and pharmacological approaches, we identified upregulated $K_V$ channel current density in cerebral arterial myocytes as the heretofore-unrecognized downstream effector of this TIMP3-sensitive pathway by which excess TIMP3 reduces arterial tone in the $TgNotch3^{R169C}$ CADASIL model. Collectively, these data suggest that elevated TIMP3 blunts the activity of the ADAM17/HB-EGF/(ErbB1/ErbB4) pathway in cerebral arterial myocytes, thereby attenuating myogenic responses in brain arteries and compromising CBF regulation in CADASIL (*Figure 8*).

Our data provide the first evidence for a mechanistic link between a change in a component of the extracellular matrix of cerebral arteries—TIMP3—and a pathogenic alteration in the density of an ion channel—$K_V$—in cerebral arterial myocytes. $K_V$ channels are powerful negative regulators of arterial tone, which act by exerting a tonic hyperpolarizing influence on the membrane potential of arterial smooth muscle cells that serves to limit pressure-induced depolarization and vasoconstriction (*Longden et al., 2015*). Our results introduce the novel concept that the concentration of TIMP3 in brain vessels regulates arterial tone and blood flow by playing a critical role in adjusting $K_V$ channel density. We surmise that such an extracellular matrix-dependent paradigm may be at play in more

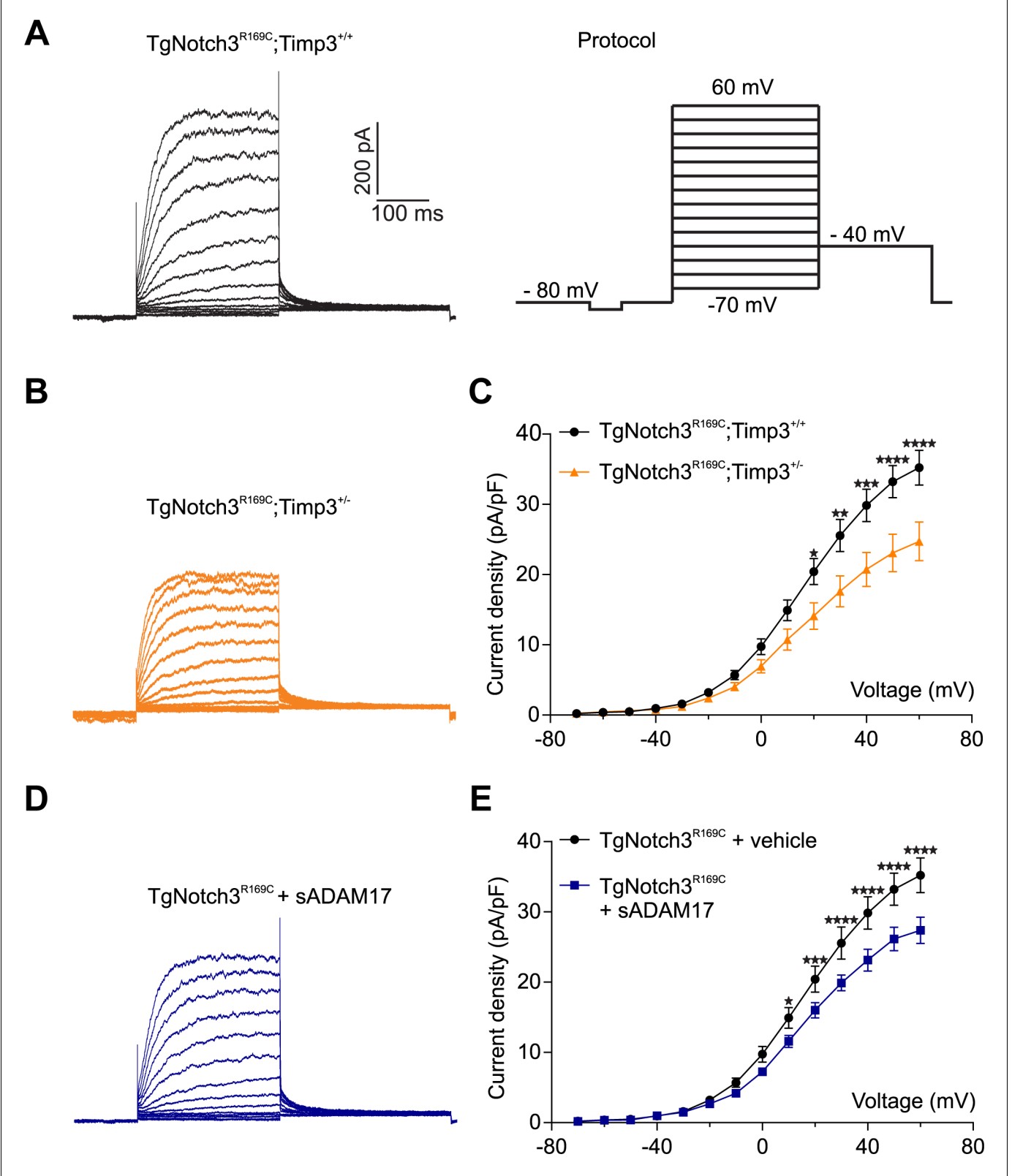

**Figure 7.** TIMP3 haploinsufficiency and exogenous sADAM17 decrease $K_V$ channel current density in cerebral smooth muscle cells from *TgNotch3^{R169C}* mice. (A,B) Typical family of $K_V$ currents recorded in isolated cerebral smooth muscle cells from double-mutant *TgNotch3^{R169C};Timp3^{+/-}* mice, with *Timp3* haploinsufficiency in the context of *Notch3^{R169C}* overexpression (B), and *TgNotch3^{R169C};Timp3^{+/+}* mice, with wild-type *Timp3* in the context of *Notch3^{R169C}* overexpression (A) elicited by voltage pulses from −70 mV to +60 mV in the presence of 1 μM paxilline (included to block BK channel

*Figure 7 continued on next page*

*Figure 7 continued*

currents). (C) Summary of current density results, showing that current density is decreased in myocytes of *TgNotch3$^{R169C}$;Timp3$^{+/-}$* mice compared with those of *TgNotch3$^{R169C}$;Timp3$^{+/+}$* mice. (D) Typical family of K$_V$ currents recorded in isolated cerebral smooth muscle cells from *TgNotch3$^{R169C}$* mice incubated with soluble ADAM17 (3.2 nM). (E) Summary of current density results, showing that the current density of *TgNotch3$^{R169C}$* mice is decreased in the presence of sADAM17. Significance was analyzed by two-way repeated measures ANOVA followed by Bonferroni post-hoc test (n = 7–8 cells/group; 1 cell/mouse). Error bars indicate SEM.

The following source data and figure supplements are available for figure 7:

**Source data 1.** Comparison of cerebral K$_V$ current properties.
**Source data 2.** Numerical data that were used to generate the graphs in *Figure 7*.
**Figure supplement 1.** Exogenous TIMP3 increases voltage-gated potassium (K$_V$) channel current density in cerebral smooth muscle cells.
**Figure supplement 2.** Analyses of cerebral K$_V$ current properties.

common forms of cerebral small vessel disease where remodeling of the vascular extracellular matrix is a key feature (*Joutel et al., 2016*).

Our results indicate that, under physiological conditions, tonic activity of the ADAM17/HB-EGF/(ErbB1/ErbB4) pathway prevents excess accumulation of K$_V$ channels at the plasma membrane and thereby maintains myogenic tone and robust CBF responses to neural activity and vasodilators. Interestingly, we found that only factors that inhibit this pathway had a functional effect; activating this pathway in wild-type mice by providing sADAM17 or sHB-EGF did not enhance evoked CBF responses or myogenic tone. This suggests that the set point of this pathway in a physiological in vivo setting is already at maximum. In support of this interpretation is a recent study showing that genetic overexpression of ADAM17 protein does not result in enhanced shedding activity in vivo (*Yoda et al., 2013*). On the other hand, decreasing K$_V$ current density in cerebral artery myocytes, which is at least 50% lower than that in peripheral artery myocytes (*Dabertrand et al., 2015*), could be an in vivo rate-limiting step following physiological activation of this pathway. Notably however, studies in experimental models of aneurysmal subarachnoid hemorrhage indicate that this pathway can be further activated in a pathological context. Indeed, Wellman and colleagues have shown that the blood component, oxyhemoglobin, causes suppression of K$_V$ currents in cerebral arterial myocytes through HB-EGF–mediated activation of ErbB1/ErbB4, resulting in membrane depolarization and enhanced tone of brain arteries (*Nystoriak et al., 2011*).

In the present study, we found that any genetic or pharmacological maneuver that blocked the ADAM17/HB-EGF/(ErbB1/ErbB4) pathway attenuated both the myogenic tone of brain arteries and the increase in CBF responses evoked by diverse stimuli; conversely, exogenous sADAM17 and sHB-EGF could overcome the reduction in both myogenic tone and evoked CBF responses elicited by the R169C Notch3 mutation or elevated TIMP3. Moreover, our previous (*Dabertrand et al., 2015*) and current results collectively indicate that upregulation of K$_V$ channels is sufficient to explain the decrease in myogenic tone, without an involvement of the endothelium or large conductance, voltage and Ca$^{2+}$ activated K$^+$ (BK) channels. Nonetheless, we cannot exclude an effect on other channels engaged by pressure. Given the key role of K$_V$ channels in the regulation of arterial tone, these findings are consistent with the interpretation that the smooth muscle ADAM17/HB-EGF/(ErbB1/ErbB4)/K$_V$ pathway regulates evoked CBF responses by elevating the physiological tone of brain arteries, and that the reduction in myogenic tone caused by inhibition of this pathway by excess TIMP3 in the extracellular matrix surrounding smooth muscle cells likely accounts for the attenuation of evoked CBF responses in CADASIL (*Figure 8*). A previous study in acute brain slices provides additional support for this interpretation, showing that the initial degree of arteriolar tone determines the diameter changes elicited by functional hyperemia (*Blanco et al., 2008*). On the other hand, a transient loss of myogenic tone is expected to increase resting CBF, and vice versa. Unexpectedly, we found that acute pharmacological blockade of the ADAM17/HB-EGF/(ErbB1/ErbB4) pathway did not affect resting CBF (or inconsistently increased it), despite its ability to profoundly reduce myogenic responses of brain arteries ex vivo. Also, neocortical application of exogenous

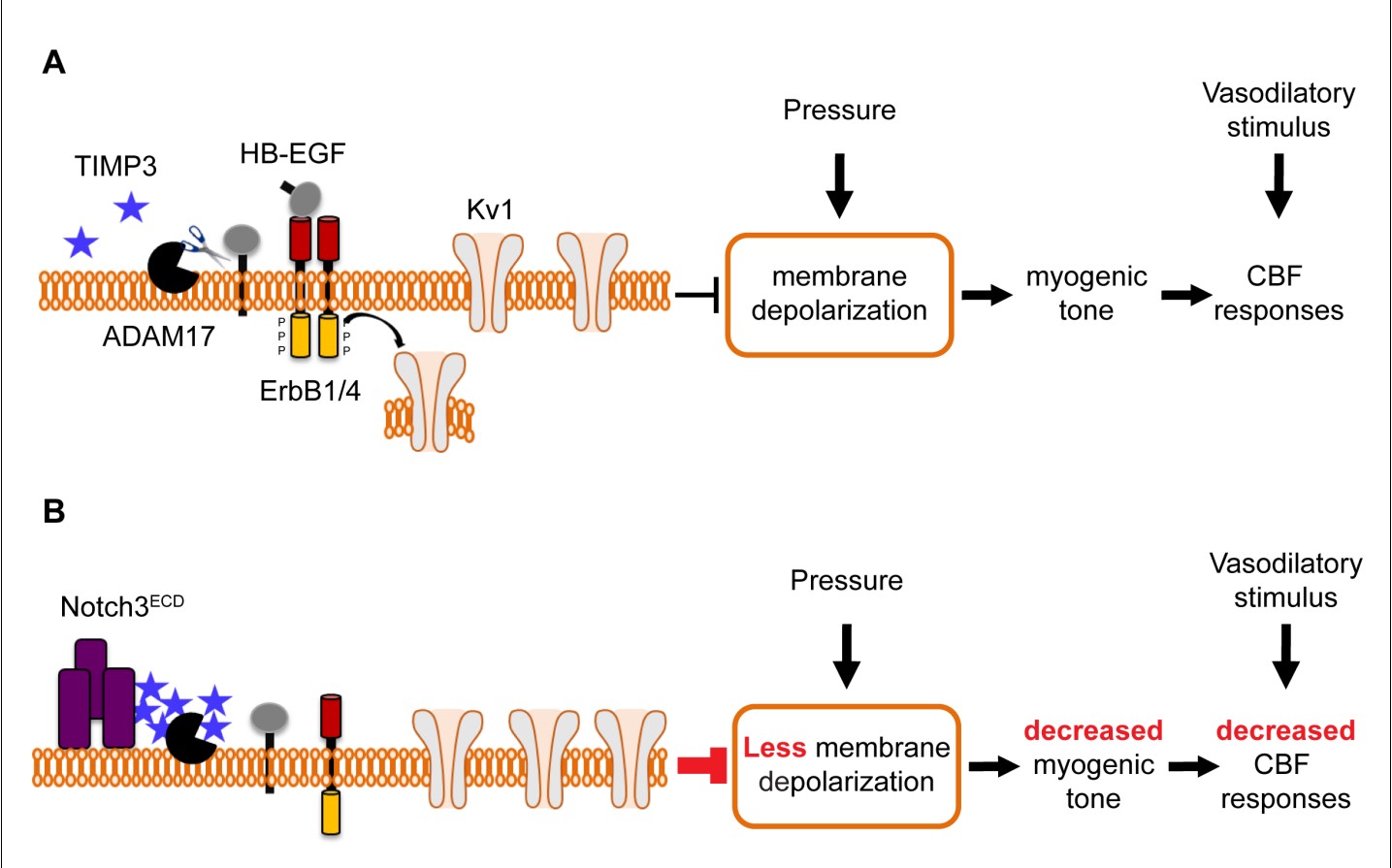

**Figure 8.** Proposed model of TIMP3 regulation of cerebral arterial tone and CBF responses. (**A**) Under physiological conditions (upper panel), TIMP3 is present in a low abundance in the extracellular matrix of brain arteries. ADAM17 at the cell surface of cerebral arterial myocytes is therefore active and able to cleave and release sHB-EGF, resulting in ErbB1/ErbB4 activation and $K_V1$ channel endocytosis. The internalization of $K_V1$ channels relieves the tonic hyperpolarizing influence of these channels on the membrane potential of arterial myocytes, thereby allowing full development of pressure-induced vasoconstriction (myogenic tone) of brain arteries and enabling full CBF responses to whisker stimulation and vasodilators. (**B**) In CADASIL (lower panel), Notch3ECD accumulates at the surface of smooth muscle cells, leading to an increase in the amount of TIMP3, which binds to and inhibits ADAM17, blunting sHB-EGF release and ErbB1/ErbB4 activity, and thereby decreasing $K_V1$ endocytosis. The resulting increase in $K_V1$ current density hyperpolarizes arterial myocytes, acting as a brake to limit the development of myogenic tone and evoked CBF responses.

sADAM17 unexpectedly increased resting CBF in *TgNotch3R169C*mice, despite its ability to increase and normalize myogenic tone in these mice. However, although myogenic tone and myogenic responses are known to contribute to the regulation of resting CBF, their relative importance are hard to quantify and poorly understood; their contribution may also change depending on conditions or disease states and other mechanisms —metabolic, neural, endothelial—also influence or contribute to resting CBF (*Cipolla, 2009*). It is also possible that overall resting CBF does not change despite seeing a change in myogenic tone in one portion of the vasculature because of compensatory adjustments in vessels downstream. Simultaneous in vivo recordings of blood flow and vessel diameters may be of interest to address this possibility. Moreover, it should be stressed that in our experiments cell populations targeted by pharmacological compounds likely differ depending on whether the compound is topically applied in vivo over the somatosensory cortex or incubated ex vivo with isolated brain arteries. In particular, proteins or peptides topically applied in vivo are thought to target only the abluminal surface of the vessel (smooth muscle cells) (*Park et al., 2013*) and may target other brain cells (e.g., astrocytes), whereas ex vivo incubation targets only vascular cells, including both abluminal and luminal (endothelial cell) surfaces. In this regard, involvement of the ADAM17/HB-EGF/(ErbB1/ErbB4) pathway in cells other than arterial myocytes cannot be ruled

out. Finally, our results may point toward the involvement of other downstream effectors in addition to $K_V$ channels in the regulation of CBF by this pathway.

The fact that increasing TIMP3 or decreasing ADAM17 caused a concentration-dependent impairment of cerebrovascular function taken together with the observation that exogenous sADAM17 is capable of overcoming elevated TIMP3-induced cerebrovascular dysfunction indicates that ADAM17 activity depends on the relative activity of ADAM17 and TIMP3 in brain arteries. Several lines of evidence from cell systems indicate that the bulk of ADAM17 is intracellular, whereas the majority of ADAM17 shedding activity occurs at the cell surface, where ADAM17 can associate with its natural inhibitor TIMP3 (*Xu et al., 2012*; *Chapnick et al., 2015*). Thus, the ratio of TIMP3 and ADAM17 at the cell surface is likely a key determinant of ADAM17 activity. Biochemical confirmation of this in brain arteries will require further investigation, although the lack of ADAM17 and TIMP3 antibodies suitable for immunohistochemistry, the tiny amount of material provided by segments of cerebral arteries for biochemical studies, and the lack of in situ or specific assay to assess ADAM17 activity in tissues remain major technical obstacles.

Although many of the molecular details of the mechanism responsible for EGFR-mediated suppression of $K_V$ channels in cerebral arterial myocytes remain unsettled, previous studies have shown that activation of EGFR tyrosine kinase activity can suppress $K_V$ channel activity through enhanced endocytosis (*Koide et al., 2007*; *Ishiguro et al., 2006*). Functional homo- or heteromeric $K_V$ channels are formed from four α-subunits, plus additional β-subunits. $K_V1.5$, and to a lesser extent $K_V1.2$, are the predominant α-subunits in rodent brain arteries (*Thorneloe et al., 2001*; *Straub et al., 2009*). Whereas direct tyrosine phosphorylation of the channel has been identified as the mechanism regulating $K_V 1.2$ endocytosis in HEK or neuronal cells (*Nesti et al., 2004*), a role for this mechanism in $K_V1.5$ endocytosis has not yet been demonstrated (*Ishiguro et al., 2006*). Whether other subunits within the $K_V 1.5$ channel complex or a closely associated protein is the target of phosphorylation remains to be tested. On the other hand, $K_V$ channel suppression could be mediated by enhanced lysosomal or proteasomal degradation, as recently shown for $K_V 1.5$ in mesenteric arteries (*Kidd et al., 2015*). Future experiments are needed to elucidate mechanisms responsible for the regulation and trafficking of $K_V1$ channels in cerebral arterial myocytes.

In summary, our study has uncovered a novel and central role for the ADAM17/HB-EGF/ErbB/$K_V$ signaling pathway in the physiological and pathological control of CBF and arterial tone. Our results highlight a heretofore-unrecognized mechanistic link between pathological alterations of the vascular extracellular matrix and $K_V$ channel density that underlies cerebrovascular dysfunction in CADASIL. We believe that this novel extracellular matrix-dependent mechanism establishes an important paradigm for cerebrovascular regulation. Importantly, illumination of its dysfunction in cerebral small vessel disease offers multiple points of potential therapeutic intervention that may prove to be more easily druggable than pathological changes in vascular extracellular matrix.

## Material and methods

### Reagents

Acetylcholine, adenosine, the selective ErbB1/ErbB4 inhibitor tyrphostin AG1478, and the selective ErbB2 inhibitor tyrphostin AG825 (*Levitzki and Gazit, 1995*) were purchased from Sigma Aldrich (St. Louis, MO). Heparin was purchased from Merck Millipore (Molsheim, France). The ADAM inhibitors GI254023X (ADAM 10) and GW413333X (ADAM10/ADAM17) (*Hundhausen et al., 2003*) were synthesized by Iris Biotech (Marktredwitz, Germany). Murine recombinant TIMP1, murine soluble ErbB1, ErbB3 and ErbB4 receptor traps (ErbB1-IgG1 Fc, ErbB3-IgG2 Fc and ErbB4-IgG2 Fc), and control IgG1 Fc and IgG2 Fc fragments, as well as human bioactive ADAM17 were purchased from R&D Systems (Lille, France). Murine recombinant TIMP2 and TIMP3 were purchased from Uscn Life Science (Houston, TX, and murine sHB-EGF was purchased from BioVision (Milpitias, CA). The p21 peptide (H-KKK KKG KGL GKK RDP CLR KYK-OH), which competitively inhibits HB-EGF binding to heparan sulfate proteoglycan (*Higashiyama et al., 1993*), and a mutated, inactive peptide in which all lysine residues that are important for inhibitory activity are replaced with alanines (p21-mut; H-AAA AAG AGL GAA RDP CLR AYA-OH) were purchased from Eurogenetec (Seraing, Belgium) and resuspended at a concentration of 25 mM in DMSO, following the manufacturer's directions. Paxilline was purchased from A.G. Scientific (San Diego, CA). Apamin and charybdotoxin were

purchased from Enzo Life Sciences (Farmingdale, NY). Papain and collagenase type 4 were purchased from Worthington Biochemical Corporation (Lakewood, NJ). All other chemicals were obtained from Sigma Aldrich.

## Mice

*Experiments were conducted in* FVB/N mice (Charles River Laboratories, France); transgenic mice overexpressing the R169C mutation of Notch3 (*TgNotch3*$^{R169C}$, line 88), bred on an FVB/N background (*Joutel et al., 2010*); transgenic mice overexpressing human TIMP3 (*TgBAC-TIMP3*), bred on a hybrid background (88% FVB/N/12% C57Bl/6) (*Capone et al., 2016*); homozygous *Adam17*$^{ex/ex}$ mice, which express profoundly reduced ADAM17 protein levels in all tissues; and heterozygous *Adam17*$^{ex/+}$ mice and wild-type littermates (*Chalaris et al., 2010*), maintained on a C57BL/6-SV129 hybrid background. Genotyping analyses were performed by polymerase chain reaction (PCR) using the following primer pairs: *TgNotch3*, 5'-TCA ACG CCT TCT CGT TCT TC-3' (forward) and 5'-AAT ACC GTC GTG CTT TCG AG-3' (reverse); *TgBAC-TIMP3*, 5'-CCA GGA GAC AGC AAG TAG CC-3' (forward) and 5'-GCT GCT GTT TAG GGA TCT GC-3' (reverse); *Adam17* mutant and wild-type allele, 5'- TAT GTG ATA GGT GTA ATG -3' (forward) and 5' CTT ATT ATT CTC GTG GTC ACC - 3'(reverse).

Mice were bred and housed in pathogen-free animal facilities and fed a standard diet ad libitum with free access to water. All experiments described in this study were conducted in full accordance with the guidelines of our local Institutional Animal Care and Use Committee (Lariboisière-Villemin), with every effort being made to minimize the number of animals used. All mice were male, aged 2 months, except for *TgNotch3*$^{R169C}$, *TgBAC-TIMP3* and non-transgenic littermate mice, which were 6 months old. We report this study in compliance with the ARRIVE guidelines.

## Western blotting

Protein extracts were prepared from cerebral pial arteries and immunoblotted using rabbit polyclonal anti-ADAM17 (18.2) (1:2000) (*Chalaris et al., 2010*) and anti-smooth muscle α-actin (Clone 1A4, Dako; Les Ulis, France) antibodies, as previously described (*Monet-Leprêtre et al., 2013*). Densitometric quantification of band intensity was performed using ImageJ (version 10.2, NIH).

## In vivo analysis of cerebrovascular reactivity

### Surgical procedure

Mice were anesthetized with isoflurane (maintenance, 2%), tracheally intubated, and artificially ventilated with an oxygen-nitrogen mixture using a ventilator (Sar-830/P; CWE Inc.). The femoral artery was cannulated for recording mean arterial pressure and collecting blood samples. A small craniotomy (2 × 2 mm) was performed to expose the whisker-barrel area of the somatosensory cortex, the dura was removed, and the site was superfused with Ringer's solution (37°C, pH 7.3–7.4). After surgery, isoflurane was gradually discontinued and anesthesia was maintained with urethane (750 mg kg$^{-1}$) and chloralose (50 mg kg$^{-1}$). Rectal temperature was maintained at 37°C, and arterial blood gases were measured. The level of anesthesia was monitored by testing corneal reflexes and motor responses to tail pinch. To minimize confounding effects of anesthesia on vascular reactivity, we kept the time interval between the administration of urethane-chloralose and the testing of CBF responses consistent among the different groups of mice studied. Arterial blood pressure, blood gases, and rectal temperature were monitored and controlled.

### CBF monitoring

Relative CBF was continuously monitored at the site of the cranial window using a laser-Doppler probe (Moor Instruments; Axminster, UK) positioned stereotaxically 0.5 to 1 mm from the cortical surface. CBF values were expressed as percent increase relative to the resting level [(CBF$_{stimulus}$–CBF$_{resting}$)/CBF$_{resting}$]. Zero values for CBF were obtained after the heart was stopped by an overdose of isoflurane at the end of the experiment (*Capone et al., 2012*).

CBF recordings were started after arterial pressure and blood gases had reached a steady state, as previously described (*Capone et al., 2012*). All pharmacological agents and drugs studied were dissolved in a modified Ringer's solution (*Girouard et al., 2006*). The increase in CBF produced by somatosensory activation was assessed by stimulating the whiskers contralateral to the cranial

window by side-to-side deflection for 60 s. The endothelium-dependent vasodilator acetylcholine (10 µmol/L; Sigma-Aldrich) was topically superfused for 5 min, and the resulting changes in CBF were monitored. CBF responses to the smooth muscle-dependent relaxant adenosine (400 µM; Sigma-Aldrich) were also examined.

### Pharmacology

The effects of drugs on cerebrovascular reactivity were examined by testing CBF responses to whisker stimulation and adenosine before superfusion, during superfusion of the cranial window with Ringers' solution containing the appropriate vehicle (first step), and after superfusion with Ringers' solution containing the drug for 30 to 90 min (second step) (*Figure 1A*). In some studies, a third step was added to test the joint effect of two compounds. Drug concentrations were based on prior reports (*Schwarz et al., 2013*), initial experiments, and a report showing that 100- to 1000-fold higher amounts of drugs are required to achieve effective concentrations in the brain in vivo (*Westerink and De Vries, 2001*). Chemical inhibitors were used at concentrations ranging from 0.1 to 20 µM, recombinant proteins were used at concentrations ranging from 8 to 70 nM, and synthetic peptides were used at 12 µM.

### Assessment of brain penetration of recombinant protein topically applied over the cranial window

Surgical procedures were performed as described above. Ringers' solution containing fluorescein isothiocyanate-labeled serum albumin (FITC-BSA) was topically superfused over the somatosensory cortex for 30 min. The mouse was then transcardially perfused with 20 ml of phosphate-buffered saline (PBS) and 30 ml of 4% paraformaldehyde, and, after sacrificing the animal, the brain was removed and post-fixed in 4% paraformaldehyde overnight. The brain was sectioned in 50-µm-thick coronal slices through the perfusion site using a vibratome, washed in PBS, immunostained with Alexa 594-conjugated anti-smooth muscle α-actin (1:500, clone 1A4; Abcam, Paris, France) and mounted on a glass slide in a drop of Dako fluorescence mounting medium (Dako; Les Ulis, France). Stained sections were imaged with a Nikon Eclipse 80i microscope (Nikon; Champigny sur Marne, France); images were captured with an Andor Neo sCMOS camera and NIS Elements BR v 4.0 software (Nikon) using identical settings across compared groups.

### Local field potential recordings

Mice were anesthetized and surgically prepared as described above. Field potentials were recorded using a stainless steel bipolar electrode placed in the somatosensory cortex contralateral to the activated whiskers (3 mm lateral and 1.5 mm caudal to bregma; depth, 0.5 mm). The somatosensory cortex was activated by two needle electrodes (21 gauge) subdermally inserted in the whisker pad. Each stimulation trial lasted for 1 min (0.65 mA; 0.5 Hz; pulse duration, 1 ms) and the interval between two trials was 10 min. Eight consecutive stimulation trials were performed on each mouse. The first four cycles were carried out in presence of vehicle, and the subsequent four trials were performed in the presence of recombinant TIMP3 (40 nM); analyses were performed on the average of four trials. Data were obtained and recorded using the MP36R System (Biopac System, CA) and analyzed off-line using AcqKnowledge Software (Biopac System, CA).

### Pharmacology on pressurized brain arteries and parenchymal arterioles

After overdosing with $CO_2$, mice were decapitated and their brains were harvested. Arterial segments of the posterior cerebral artery and precapillary segments of parenchymal arterioles that arise from the middle cerebral artery M1 region and perfuse the neocortex were dissected, cannulated on two glass micropipettes in an organ chamber containing physiological salt solution (PSS) maintained at 37°C (pH 7.4), and pressurized using an arteriograph system (Living Systems Instrumentation, Inc., St. Albans, VT) as previously described (*Joutel et al., 2010*). Once prepared, arteries were allowed to stabilize for at least 60 min at a pressure of 60 mmHg until the development of basal tone. Pressure was then switched to 20 mmHg and compounds were added to the chamber for 20 to 60 min before increasing the intraluminal pressure to 40, 60, 80 and 100 mm Hg using a pressure-servo control pump. Vessel internal diameter was continuously recorded using a CCD camera and edge-detection software (Biopac MP150; Biopac Systems Inc., CA or AcqKnowledge Software; IonOptix,

Milton, MA). Diameters measured in PSS were considered active diameters. At the conclusion of each experiment, maximal dilation was obtained in nominally $Ca^{2+}$-free PSS containing EGTA (2–5 mM; Sigma). Artery diameters are given in micrometers. Myogenic tone was expressed as the percentage of passive diameter ([passive diameter – active diameter]/passive diameter $\times$ 100).

Compound concentrations were based on initial experiments of cerebrovascular reactivity and used at approximately one fifth of the concentration used in vivo.

## Arterial myocyte isolation and electrophysiology

Anterior, middle, and posterior cerebral arteries and arterioles were cleaned of connective tissue and placed in cell-isolation solution. Single smooth muscle cells were isolated from cerebral arteries by enzymatic digestion in papain (0.5 mg/mL) and dithioerythritol (1 mg/mL) for 12 min, followed by a second digestion in collagenase type 4 (1 mg/mL) without $Ca^{2+}$ for 10 min. Digested tissue was washed out and gently triturated with a fire-polished glass pipette. The single-cell suspension of myocytes was refrigerated until use (typically 4–6 hr). Outward $K^+$ currents were recorded from single cells in the presence of 1 µM paxilline (to block BK currents) at room temperature using the perforated-cell configuration of the patch-clamp technique. Recording electrodes with resistances of 2–4 MΩ were pulled from borosilicate glass and backfilled with a pipette solution of appropriate composition. Currents were recorded from cells on an Axopatch 200B amplifier, filtered at 2 kHz using a low-pass Bessel filter, and digitized at 10 kHz (Digidata 1322A; Molecular Devices). pCLAMP-9 software (Molecular Devices) was used for data recording and analysis. The composition of cell isolation solution was 60 mM NaCl, 85 mM Na-glutamate, 3 mM KCl, 2 mM $MgCl_2$, 10 mM HEPES, 10 mM glucose, 7 mM mannitol, pH 7.4. For patch-clamp experiments, the bath solution composition was 137 mM NaCl, 3 mM KCl, 0.1 mM $CaCl_2$, 4 mM glucose, 10 mM HEPES (pH 7.3), and contained paxilline (1 µM); the pipette solution was 10 mM NaCl, 30 mM KCl, 110 mM K-aspartate, 1 mM $MgCl_2$, 10 mM HEPES (pH 7.2), and contained 250 µg/mL amphotericin B. Families of outward $K_V$ currents were elicited by series of 10-mV depolarizing steps from −70 mV to +60 mV, from a holding potential of −80 mV (*Figure 7a*). Current density was calculated by dividing membrane current amplitude at the end of the pulse by cell capacitance.

The relationship between myocyte membrane voltage and the amplitudes of tail currents (*I*) was fit to the Boltzmann equation,

$$I = \frac{I_{max}}{1 + e^{(V_{0.5}+V)/k}}$$

where $I_{max}$ is the measured peak tail current, which allows determination of the half-maximal activation potential ($V_{0.5}$) and slope (*k*).

### Estimation of the number of channels

In symmetrical, high extracellular $K^+$ ($[K^+]_o$) and intracellular $K^+$ ($[K^+]_i$) solutions, the Goldman-Hodgkin-Katz flux equation (*Hodgkin and Katz, 1949*) for $K^+$ predicts a linear relationship between channel current amplitude, *I,* and membrane potential:

$$i = P_K \times \frac{EF^2}{RT} \times \frac{[K^+]_o - [K^+]_i \cdot \exp(EF/(RT))}{1 - \exp(EF/(RT))}.$$

With $[K^+]_o = [K^+]_i$ the equation becomes

$$i = P_K \times EF^2[K^+]/(RT),$$

where $P_K$ is the permeability to $K^+$ of a single channel (in cm/s); *E* is the membrane potential (*V*); *F*, *R* and *T* have their usual meanings; and the $K^+$ concentration is given in mol/mL. Since $K^+$ single-channel conductance is defined as $\gamma = i/E$, permeability can be defined as

$$P_K = \gamma/[K^+] \times \left(RT/F^2\right).$$

Using a single-channel conductance of 15 pS recorded in inside-out patches with symmetrical $[K^+] = 140$ mM (*Aiello et al., 1998*), we calculated $P_K = 2.83 \times 10^{-14}$ cm/s at 23°C. The single-channel current amplitude, *i*, was then estimated using $P_K$ and the Goldman-Hodgkin-Katz flux equation at a

given voltage ($-40$ mV) with $[K^+]_o$ = 3 mM and $[K^+]_i$ = 140 mM. Finally, the number of channels ($N$) was determined using the macroscopic current amplitude ($I$) equation, $I = iNP_o$, with $P_o$ = 0.014 at $-40$ mV (*Aiello et al., 1998*).

## Statistical analysis

Data are expressed as means ± SEM. Sample size needed for CBF and myogenic tone analysis as well as for electrophysiology experiments was determined based on our prior works (*Dabertrand et al., 2015*), (*Capone et al., 2016*); n values indicate the number of biological replicates. CBF responses were analyzed by one-way analysis of variance (ANOVA) or repeated-measure ANOVA followed by Bonferroni or Tukey post-hoc tests. Evoked potential fields were analyzed using unpaired Student's t-test. Myogenic tone and current densities were analyzed by two-way repeated-measure ANOVA followed by Bonferroni post-hoc tests. All statistics were performed using Graph Pad Prism. Differences with p-values < 0.05 were considered statistically significant. The significance level was set at p<0.05.

## Acknowledgements

We thank Serge Charpak for assistance with local field potential recordings, and David Hill-Eubanks for discussions and editorial input. We thank TAAM-Orleans (Caroline Bertrand & Alexandre Diet) and University Paris Denis Diderot-site Villemin (Suzanne Orville & Frédéric Baudin) for animal housing.

## Additional information

### Funding

| Funder | Grant reference number | Author |
|---|---|---|
| United Leukodystrophy Foundation | CADASIL Research Grant | Fabrice Dabertrand |
| Deutsche Forschungsgemeinschaft | DFG, SFB877 project A1 | Athena Chalaris Stefan Rose-John |
| Deutsche Forschungsgemeinschaft | The Cluster of Excellence Inflammation at Interfaces | Athena Chalaris Stefan Rose-John |
| Fondation Leducq | Transatlantic Network of Excellence on the Pathogenesis of Small Vessel Disease of the Brain | Mark T Nelson Anne Joutel |
| European Union | Horizon 2020 research and innovation programme SVDs@target, grant agreement No 666881 | Mark T Nelson Anne Joutel |
| National Institutes of Health | R37DK053832 | Mark T Nelson |
| Totman Medical Research Trust | | Mark T Nelson |
| National Institutes of Health | PO1HL095488 | Mark T Nelson |
| National Institutes of Health | RO1HL44455 | Mark T Nelson |
| National Institutes of Health | R01HL121706 | Mark T Nelson |
| National Institutes of Health | R01HL131181 | Mark T Nelson |
| Agence Nationale de la Recherche | ANR Genopath 2009-RAE09011HSA | Anne Joutel |
| Agence Nationale de la Recherche | ANR Blanc 2010-RPV11011HHA | Anne Joutel |

The funders had no role in study design, data collection and interpretation, or the decision to submit the work for publication.

## Author contributions

CC, Participated in the design of the study, Performed CBF experiments, Conducted local field potential recordings, Analyzed the data and Contributed to drafting of the paper; FD, Carried out myogenic tone analyses, Performed arterial myocyte electrophysiology, Analyzed the data and Contributed to drafting of the paper; CB-M, Carried out myogenic tone analyses, Analyzed the data and Contributed to drafting of the paper; AC, SS, Provided critical reagents and mice and participated to drafting of the paper; LG, VD-D, CH, Contributed to acquisition and analysis of data, and to drafting of the paper; SR-J, Provided critical reagents and mice and participated to drafting and revising the paper; MTN, Participated in the design of the study and data analysis, Wrote the manuscript; AJ, Conceived the study, Supervised the project, Analyzed the data and Wrote the manuscript.

## Author ORCIDs

Anne Joutel, http://orcid.org/0000-0003-3963-3860

## Ethics

Animal experimentation: All experiments were conducted in full accordance with the French guidelines for the use of animals in research and were approved by the "Lariboisière-Villemin" Institutional Animal Care and Use Committee (C2EA 09), with every effort made to minimize the number of animals used.

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
