## [Decision Letter]

Thank you for submitting your article "Mechanistic insights into a TIMP3-sensitive pathway constitutively engaged in the regulation of cerebral hemodynamics" for consideration by *eLife*. Your article has been favorably evaluated by a Senior editor and three reviewers, including John P Adelman (Reviewer #2) Scott Earley (Reviewer #3) and David E Clapham (Reviewer #1), who is a member of our Board of Reviewing Editors.

The reviewers have discussed the reviews with one another and the Reviewing Editor has drafted this decision to help you prepare a revised submission.

Summary:

The manuscript from Joutel and colleagues reports results that examine the mechanisms by which TIMP3 regulates cerebral arterial tone and CBF responses. Previous work had shown that TIMP3 abnormally accumulates in the extracellular matrix of brain vessels of patients and *TgNotch3^R169C^* mice with CADASIL, a monogenic form of small vessel disease. Genetic overexpression of TIMP3 recapitulates both CBF and myogenic response deficits of the CADASIL model, while genetic reduction of TIMP3 restores normal function. Also, they had previously shown that downregulation of plasma membrane K_v_ channels restored normal responses to pressure. Here, they demonstrate that TIMP3 mediates cerebrovascular function through ADAM17/HB-EGF/EGFR (ErbB1/ErbB4) signaling pathway. In CADASIL, accumulation of TIMP3 leads to the inhibition of ADAM17, consequently, prevents sHB-EGF from being cleaved and affects the subsequent ErbB1/ErbB4 activity. Through a mechanism yet to be determined, this apparently decreases K_v_ 1 channel endocytosis. The enhanced K_v_ 1 channel activity due to increased current density limits the myogenic tone and evoked CBF responses.

This is a very well written manuscript, the data are clearly presented and the results are significant.

Revisions to address:

1) Application of sADAM17 (16 nM) did not affect responses in wt. This suggests that endogenous levels of ADAM17 are saturated. Was a higher concentration tested? Also, what was the effect of exogenous application of sADAM17 on the stronger hypomorph, *ex/ex*? This would be interesting as it would determine whether the signaling pathway remains intact even when ADAM17 is largely abolished. Indeed, even in the genetic model of TIMP3 over-expression, ADAM17 application improved CBF responses.

2) Throughout the manuscript, using the in vivo model of direct cortical application, the responsiveness to the various treatments and genetic models are presented as relative, percent changes. Since the system seems to operate at maximum in the basal condition, it would be interesting to know whether there are significant differences in the absolute sense. For example, is resting flow the same for the various genetic models? Does application of ADAM17 normalize CBF in the hypomorph to the same absolute value as wt?

3) Application of ADAM17 to the hypomorph did not affect the relative change in resting flow, but increased it in the N3^R169C^ model. If the effect of both of these models is to reduce the effective ADAM17 activity, why are the responses on resting flow different?

4) The main conclusion of this manuscript is that TIMP3 blunts the ADAM17/HB-EGF/ErbB/K_v_ pathway to disrupt neurovascular coupling (NVC) by diminishing myogenic constriction. The authors posit that because arteries and arterioles are already in a dilated state, intrinsic NVC coupling mechanisms cannot further increase CBF. If this is the case, why are increases in CBF due to loss of myogenic tone not seen when exogenous TIMP3 is acutely applied (i.e., Figure 1)?

5) The authors show that administration of sADAM17 restores NVC in *ADAM17^ex/wt^*mice (Figure 2). It would be useful to show that sADAM17 also restores impaired myogenic reactivity in isolated cerebral arteries from these mice. These new experiments would provide a direct mechanistic link between restoration of myogenic tone and repair of NVC.

6) Ex vivo experiments examining myogenic reactivity used cerebral pial arteries, but NVC is primarily regulated by parenchymal (penetrating) arterioles. Prior work from some of the co-authors and other labs indicate that the signaling pathways responsible for vascular contractility differ between smooth muscle cells from pial versus parenchymal arterioles. Additional experiments using isolated cerebral parenchymal arterioles would be helpful to resolve this issue.

7) Patch-clamp experiments – The effects of exogenous TIMP3 are quite modest and in all experiments shown here, K_v_ current densities differ only at very positive membrane potentials. The prior study from this research group (PMID:25646445) shows increases in current density between *TgNotch3^R169C^* and wild-type mice at membrane potentials that are physiologically relevant for smooth muscle cells (i.e., -30 mV). If the authors' proposed mechanism is correct, a larger influence of TIMP3 at these membrane potentials would be expected. These findings call into question the authors' contention that increases in K_v_ channel surface expression accounts for loss of myogenic contractility. Perhaps a more through characterization of electrophysiological properties is needed that would encompass other K^+^ channels (BK) as well as depolarizing influences (TMEM16A, TRPC6, TRPM4). At a minimum, the authors should at least discuss the limitations of the data that are presented.

---

## [Author Response]

*Revisions to address:*

*1) Application of sADAM17 (16 nM) did not affect responses in wt. This suggests that endogenous levels of ADAM17 are saturated. Was a higher concentration tested? Also, what was the effect of exogenous application of sADAM17 on the stronger hypomorph, ex/ex? This would be interesting as it would determine whether the signaling pathway remains intact even when ADAM17 is largely abolished. Indeed, even in the genetic model of TIMP3 over-expression, ADAM17 application improved CBF responses.*

We did not test a higher concentration of sADAM17 in wildtype mice. Interestingly, results of a recent study support the idea that endogenous levels of ADAM17 are indeed saturated. Yoda and colleagues recently showed that ADAM17 transgenic mice, which overexpress ADAM17 mRNA and protein, exhibit no overt defects and no enhanced shedding activity of ADAM17, both in vitro and in vivo (Yoda M et al., PLoS One 2013;8(1):e54412. PMID: 23342154). We thank the reviewers for this insightful comment and this point has been added in the revision (Discussion, fourth paragraph).

We acknowledge that it would have been desirable to perform more experiments in the *ADAM17^ex/wt^*and ADAM17^ex/ex^ mice. However, because we had a limited access to these mice – we did not set up the colony but imported the mice into the Institute where the CBF experiments have been conducted – and because only one compound can be tested in a given mouse, we prioritized the use of *ADAM17^ex/wt^*mice to test the effect of sADAM17, as shown on Figure 2, and the use of ADAM17^ex/ex^ mice to test the effect of sHB-EGF. Results of this later experiment, now included in the revision, showed that exogenous application of sHB-EGF significantly improved the CBF responses in the ADAM17 ^ex/ex^ mice, suggesting that the signaling pathway remains intact in these mice (Results, subsection “HB-EGF and EGFR operate downstream of ADAM17 to regulate CBF responses”, last paragraph; Figure 4—figure supplement 2).

*2) Throughout the manuscript, using the in vivo model of direct cortical application, the responsiveness to the various treatments and genetic models are presented as relative, percent changes. Since the system seems to operate at maximum in the basal condition, it would be interesting to know whether there are significant differences in the absolute sense. For example, is resting flow the same for the various genetic models? Does application of ADAM17 normalize CBF in the hypomorph to the same absolute value as wt?*

We thank the reviewers for raising this important point. Although laser Doppler flow (LDF) derived measurements provide an accurate assessment of relative CBF changes, it is usually considered that LDF is less accurate for obtaining absolute flow measurements and therefore we did not incorporate these values in the original submission. Having said that, we had an excellent day-to-day reproducibility of CBF measurements and we fully agree that absolute flow measurements in the genetic models used in the current study are worth considering. Interestingly, we found that in the *ADAM17^ex/wt^*mice, resting flow was not significantly decreased and unchanged upon topical application of sADAM17. In contrast, in TgBAC-TIMP3 and *TgNotch3^R169C^* mice, resting CBF was significantly reduced, and significantly improved upon application of sADAM17, with values close to values observed in WT mice. Therefore, it appears that application of sADAM17 in these genetic models normalize resting and evoked CBF responses towards the same absolute values of WT mice.

Accordingly, we now provide in the revision absolute values of resting flows for the genetic models (Results, subsections “ADAM17 is required for TIMP3-induced attenuation of CBF responses” and “Exogenous sADAM17 and exogenous sHB-EGF rescue CBF and myogenic-response deficits in the *TgNotch3^R169C^* CADASIL model”; Figure 2—figure supplement 2; revised Figure 6—figure supplement 1; revised [Supplementary-material SD6-data] and [Supplementary-material SD16-data]).

*3) Application of ADAM17 to the hypomorph did not affect the relative change in resting flow, but increased it in the* N3^R169C^
*model. If the effect of both of these models is to reduce the effective ADAM17 activity, why are the responses on resting flow different?*

We thank the reviewers for stressing this point. The responses on resting flow appear different in the *ADAM17^ex/wt^*and *TgNotch3^R169C^* mice if we consider the relative changes. However, results appear more consistent if we consider the absolute values. Indeed, as mentioned above, resting flow is unchanged in *ADAM17^ex/wt^*and unaffected by topical application of sDAM17. In contrast, resting flow, which is decreased in the *TgNotch3^R169C^* mice, is improved by exogenous sADAM17 towards wildtype values, suggesting that sADAM17 acts to normalize CBF responses to values observed in WT mice. We have clarified this point in the revision (Results subsections “ADAM17 is required for TIMP3-induced attenuation of CBF responses” and “Exogenous sADAM17 and exogenous sHB-EGF rescue CBF and myogenic-response deficits in the *TgNotch3^R169C^* CADASIL model”; Figure 2—figure supplement 2; revised Figure 6—figure supplement 1; revised [Supplementary-material SD6-data] and [Supplementary-material SD16-data]).

*4) The main conclusion of this manuscript is that TIMP3 blunts the ADAM17/HB-EGF/ErbB/*K_v_
*pathway to disrupt neurovascular coupling (NVC) by diminishing myogenic constriction. The authors posit that because arteries and arterioles are already in a dilated state, intrinsic NVC coupling mechanisms cannot further increase CBF. If this is the case, why are increases in CBF due to loss of myogenic tone not seen when exogenous TIMP3 is acutely applied (i.e., Figure 1)?*

We fully agree that a transient reduction in myogenic tone was theoretically expected to increase resting CBF, and vice versa. Although we proposed several explanations in the initial version of our manuscript, we acknowledge that this important issue deserves further discussion. Accordingly, we have added the following paragraph in the Discussion: “However, although myogenic tone and myogenic responses are known to contribute to the regulation of resting CBF, their relative importance are hard to quantify and poorly understood; their contribution may also change depending on conditions or disease states and other mechanisms—metabolic, neural, endothelial—also influence or contribute to resting CBF (Cipolla, 2009). It is also possible that overall resting CBF does not change despite seeing a change in myogenic tone in one portion of the vasculature because of compensatory adjustments in vessels downstream. Simultaneous in vivo recordings of local blood flow and vessel diameter may be of interest to address this possibility”.

*5) The authors show that administration of sADAM17 restores NVC in ADAM17^ex/wt^ mice (Figure 2). It would be useful to show that sADAM17 also restores impaired myogenic reactivity in isolated cerebral arteries from these mice. These new experiments would provide a direct mechanistic link between restoration of myogenic tone and repair of NVC.*

As suggested by the reviewers, these experiments have been carried out. Actually, we found that sADAM17 also restores myogenic tone in *ADAM17^ex/wt^*mice. These data have been added in the revision (Results, subsection “The ADAM17/HB-EGF/(ErbB1/ErbB4) signaling module regulates pressure-induced myogenic tone in brain arteries”; Figure 5—figure supplement 2).

*6) Ex vivo experiments examining myogenic reactivity used cerebral pial arteries, but NVC is primarily regulated by parenchymal (penetrating) arterioles. Prior work from some of the co-authors and other labs indicate that the signaling pathways responsible for vascular contractility differ between smooth muscle cells from pial versus parenchymal arterioles. Additional experiments using isolated cerebral parenchymal arterioles would be helpful to resolve this issue.*

We thank the Reviewers for this valuable comment. First, we would like to point out that, in *TgNotch3^R169C^* mice, we previously established that myogenic responses are similarly impaired in parenchymal arterioles and pial arteries (Dabertrand et al., Proc. Natl. Acad. Sci. 2015;112(7):E796–805), suggesting that the pathway altered by the CADASIL mutation is the same in the pial arteries and intraparenchymal arterioles. Second, as suggested, a new set of experiments has been carried out on parenchymal arterioles (PAs). We compared changes in the diameter of PAs in response to increasing intraluminal pressure. We observed that pre-incubation with TIMP3 blunted the myogenic response compared to pre-incubation with vehicle, whereas recombinant TIMP2 (10 nM) had no effect, suggesting that the TIMP3 sensitive pathway similarly operates in both pial and intraparenchymal arterioles. These data have been included in the revision (Results, subsection “The ADAM17/HB-EGF/(ErbB1/ErbB4) signaling module regulates pressure-induced myogenic tone in brain arteries”; Figure 5—figure supplement 1).

In addition, measurements of pressurized parenchymal arterioles are significantly more challenging than those of pial arteries, with a lower success rate (i.e. more time, money and most importantly animals sacrificed). Therefore, given the extensive number of demanding experiments in the current study and the lack of difference in myogenic impairment of parenchymal arterioles and pial arteries in response to TIMP3 as well as in CADASIL mice, we did not repeat all the pharmacological experiments, shown in the original submission, on the parenchymal arterioles.

*7) Patch-clamp experiments – The effects of exogenous TIMP3 are quite modest and in all experiments shown here,* K_v_
*current densities differ only at very positive membrane potentials. The prior study from this research group (PMID:25646445) shows increases in current density between TgNotch3^R169C^ and wild-type mice at membrane potentials that are physiologically relevant for smooth muscle cells (i.e., -30 mV). If the authors' proposed mechanism is correct, a larger influence of TIMP3 at these membrane potentials would be expected. These findings call into question the authors' contention that increases in* K_v_
*channel surface expression accounts for loss of myogenic contractility. Perhaps a more through characterization of electrophysiological properties is needed that would encompass other K^+^ channels (BK) as well as depolarizing influences (TMEM16A, TRPC6, TRPM4). At a minimum, the authors should at least discuss the limitations of the data that are presented.*

Exogenous TIMP3 increased current density by 21% at positive voltages (+ 40 mV) compared to 72% increase in myocytes from CADASIL cerebral arteries. A 21% increase in current due to an elevation of channel number would be physiological meaningful, but would not be detectable at negative voltages with our experimental approach. Our results are consistent with the idea that the mechanisms to upregulate channel number are less robust than endocytosis of K_v_ channels in isolated myocytes at room temperature. Furthermore, channel trafficking at room temperature (patch clamp analysis on cerebral arterial myocytes) is likely to be slower than at 37°C (intact pressurized arteries and in vivo).

In a previous study, we found that inhibition of BK channels had no effect on pressure-induced constrictions in WT and in CADASIL (Hannah et al., J. Cereb. Blood Flow Metab. 2011; Dabertrand et al., Circ. Res. 2012; Dabertrand et al., Proc. Natl. Acad. Sci.2015). Reduction of K_v_ channel activity by 4-AP, HB-EGF, and genetically (TIMP3^+/-^) in CADASIL to WT levels completely restored myogenic tone and in vivo CBF responses, suggesting that this mechanism was sufficient.

Nonetheless, we cannot absolutely exclude possible effects on pressure-induced hyperpolarizing and depolarizing influences on membrane potential. Therefore we have discussed this in the revision (Discussion).